# EFFICIENT AUTOMATIC GRAPH LEARNING VIA DESIGN RELATIONS

## ABSTRACT

Despite the success of automated machine learning (AutoML), which aims to find the best design, including the architecture of neural networks and hyper-parameters, conventional AutoML methods are computationally expensive and hardly provide insights into the relations of different model design choices. This work focus on the scope of AutoML on graph tasks. To tackle the challenges, we propose FALCON, an efficient sample-based method to search for the optimal model design on graph tasks. Our key insight is to model the design space of possible model designs as a *design graph*, where the nodes represent design choices, and the edges denote design similarities. FALCON features 1) a task-agnostic module, which performs message passing on the design graph via a Graph Neural Network (GNN), and 2) a task-specific module, which conducts label propagation of the known model performance information on the design graph. Both modules are combined to predict the design performances in the design space, navigating the search direction. We conduct extensive experiments on 27 node and graph classification tasks from various application domains. We empirically show that FALCON can efficiently obtain the well-performing designs for each task using only 30 explored nodes. Specifically, FALCON has a comparable time cost with the one-shot approaches while achieving an average improvement of 3.3% compared with the best baselines.

## 1 INTRODUCTION

Automated machine learning (AutoML) (Liu et al., 2019; Pham et al., 2018; Bender et al., 2018; Real et al., 2019; Zoph & Le, 2017; Cai et al., 2019; 2021; Gao et al., 2019; You et al., 2020b; Zhang et al., 2021) has demonstrated great success in various domains including computer vision (Chu et al., 2020; Ghiasi et al., 2019; Chen et al., 2019), language modeling (Zoph & Le, 2017; So et al., 2019), and recommender systems (Chen et al., 2022). It is an essential component for the state-of-the-art deep learning models (Liu et al., 2018; Baker et al., 2017; Xu et al., 2020; Chen et al., 2021).

Given a graph learning task, *e.g.,* a node/graph classification task on graphs, our goal of AutoML is to search for a model architecture and hyper-parameter setting from a design space that results in the best test performance on the task. Following previous works (You et al., 2020b), we define *design* as a set of architecture and hyper-parameter choices (*e.g.,* 3 layer, 64 embedding dimensions, batch normalization, skip connection between consecutive layers), and define *design space* as the space of all possible designs for a given task.

However, AutoML is very computationally intensive. The design space of interest often involves millions of possible designs (Elsken et al.; You et al., 2020a). Sample-based AutoML (Zoph & Le, 2017; Gao et al., 2019; Bergstra et al., 2011; Liu et al., 2017; Luo et al., 2018) has been used to perform search via sampling candidate designs from the design space to explore. One central challenge of existing sample-based AutoML solutions is its sample efficiency: it needs to train as few models as possible to identify the best-performing model in the vast design space. To improve the efficiency, existing research focuses on developing good search algorithms to navigate in the design space (White et al., 2021; Shi et al., 2020; Ma et al., 2019).

However, these methods do not consider modeling the effect of model design choices, which provides strong inductive biases in searching for the best-performing model. By "inductive bias", we refer to the patterns of multiple variables interacting together, which can happen in multiple parts of the

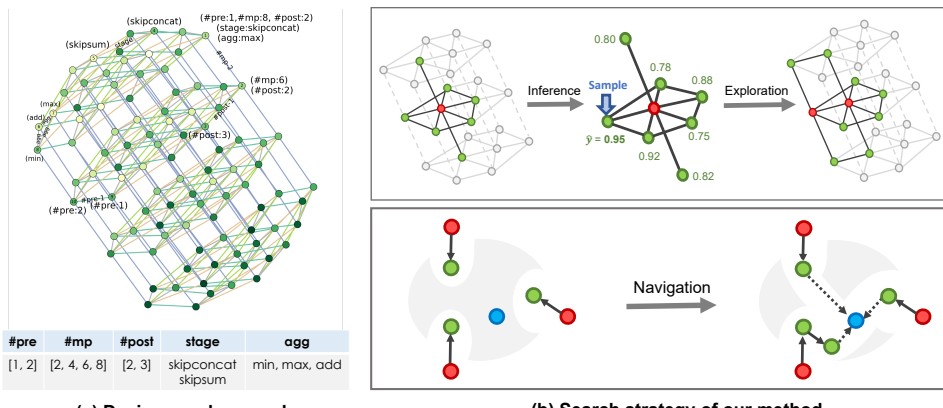

Figure 1: **Overview of FALCON**. (a) **Design graph example**. We present a small design graph on TU-COX2 graph classification dataset. The design choices are shown in the table, #pre, #mp, #post denotes the numbers of pre-processing, message passing, and post-processing layers, respectively. The better design performance, the darker node colors. (b) **FALCON search strategy.** Red: Explored nodes. Green: Candidate nodes to be sampled from. Blue: The best node. Gray: Other nodes. Locally, FALCON extends the design subgraph via a search strategy detailed in Section 3.3. Globally, FALCON approaches the optimal design navigated by the inductive bias of the design relations.

design space. Thus, an efficient search strategy should rapidly rule out a large subset of the design space with potentially bad performance leveraging such learned inductive bias.

**Proposed approach**. To overcome the limitations, we propose FALCON, an AutoML framework on graph tasks that achieves state-of-the-art sample efficiency and performance by leveraging model design insights. Our key insight is to build a *design graph* over the design space of architecture and hyper-parameter choices. FALCON extracts model design insights by learning a *meta-model* that captures the relation between the design graph and model performance and uses it to inform a sample-efficient search strategy. FALCON consists of the following two novel components.

**Design space as a graph**. Previous works view the model design space as a high-dimensional space with isolated design choices (You et al., 2020b), which offer few insights regarding the *relations* between different design choices. For example, through trial runs if we find the models with more than 3 layers do not work well without batch normalization, this knowledge can help us reduce the search space by excluding all model designs of more than 3 layers with batch normalization set to false. While such insights are hardly obtained with existing AutoML algorithms (Liu et al., 2019; Pham et al., 2018; Gao et al., 2019; Zoph & Le, 2017; Cai et al., 2019), FALCON achieves it via constructing a graph representation, design graph, among all the design choices. Figure 1(a) shows a visualization of a design graph, where each node represents a candidate design, and edges denote the similarity between the designs. See Section 3.1 for details on the similarity and graph construction.

**Search by navigating on the design graph**. Given the design graph, FALCON deploys a Graph Neural Network predictor, short for *meta-GNN*, which is supervised by the explored nodes' performances and learns to predict the performance of a specific design given the corresponding node in the design graph. The meta-GNN is designed with 1) a task-agnostic module, which performs message passing on the design graph, and 2) a task-specific module, which conducts label propagation of the known model performance information on the design graph. Furthermore, we propose a search strategy that uses meta-GNN predictions to navigate the search in the design graph efficiently.

**Experiments**. We conduct extensive experiments on 27 graph datasets, covering node- and graph-level tasks with distinct distributions. Moreover, we demonstrate FALCON' potential applicability on image datasets by conducting experiments on the CIFAR-10 image dataset. Our code is available at https://anonymous.4open.science/r/Falcon.

## 2 RELATED WORK

Automatic Machine Learning (AutoML) is the cornerstone of discovering state-of-the-art model designs without costing massive human efforts. We introduce four types of related works below.

**Sample-based AutoML methods**. Existing sample-based approaches explore the search space via sampling candidate designs, which includes heuristic search algorithms, *e.g.,* Simulated Annealing, Bayesian Optimization approaches (Bergstra et al., 2011; White et al., 2021; Ma et al., 2019), evolutionary- (Xie & Yuille, 2017; Real et al., 2017) and reinforcement-based methods (Zoph & Le, 2017; Zhou et al., 2019; Gao et al., 2019). However, they tend to train thousands of models from scratch, which results in the low sample efficiency. For example, (Zoph & Le, 2017; Gao et al., 2019) usually involve training hundreds of GPUs for several days, hindering the development of AutoML in real-world applications (Bender et al., 2018). Some hyper-parameter search methods aim to reduce the computational cost. For example, Successive Halving (Karnin et al., 2013) allocates the training resources to more potentially valuable models based on the early-stage training information. Li et al. (2017) further extend it using different budgets to find the best configurations to avoid the trade-off between selecting the configuration number and allocating the budget. Jaderberg et al. (2017) combine parallel search and sequential optimisation methods to conduct fast search. However, their selective mechanisms are only based on the model performance and lack of deep knowledge, which draws less insight into the relation of design variables and limits the sample efficiency.

**One-shot AutoML methods**. The one-shot approaches (Liu et al., 2019; Pham et al., 2018; Xie et al., 2019; Bender et al., 2018; Qin et al., 2021) have been popular for the high search efficiency. Specifically, they involve training a super-net representing the design space, *i.e.,* containing every candidate design, and shares the weights for the same computational cell. Nevertheless, weight sharing degrades the reliability of design ranking, as it fails to reflect the true performance of the candidate designs (Yu et al., 2020).

**Graph-based AutoML methods**. The key insight of our work is to construct the design space as a design graph, where nodes are candidate designs and edges denote design similarities, and deploy a Graph Neural Network, *i.e.,* meta-GNN, to predict the design performance. Graph HyperNetwork (Zhang et al., 2019a) directly generates weights for each node in a computation graph representation. You et al. (2020a) study network generators that output relational graphs and analyze the link between their predictive performance and the graph structure. Recently, Zhao et al. (2020) considers both the micro- (*i.e.,* a single block) and macro-architecture (*i.e.,* block connections) of each design in graph domain. AutoGML (Park et al., 2022) designs a meta-graph to capture the relations among models and graphs and take a meta-learning approach to estimate the relevance of models to different graphs. Notably, none of these works model the search space as a design graph.

**Design performance predictor**. Previous works predict the performance of a design using the learning curves (Baker et al., 2018), layer-wise features (Deng et al., 2017), computational graph structure (Zhang et al., 2019a; White et al., 2021; Shi et al., 2019; Ma et al., 2019; Zhang et al., 2019b; Lee et al., 2021a), or combining dataset information (Lee et al., 2021a) via a dataset encoder. To highlight, FALCON explicitly models the relations among model designs. Moreover, it leverages the performance information on training instances to provide task-specific information besides the design features, which is differently motivated compared with Lee et al. (2021b) that employs meta-learning techniques and incorporate hardware features to rapidly adapt to unseen devices. Besides, meta-GNN is applicable for both images and graphs, compared with Lee et al. (2021a).

## 3 PROPOSED METHOD

This section introduces our proposed approach FALCON for sample-based AutoML. In Section 3.1, we introduce the construction of design graph, and formulate the AutoML goal as a search on the design graph for the node with the best task performance. In Section 3.2, we introduce our novel neural predictor consisting of a task-agnostic module and a task-specific module, which predicts the performances of unknown designs. Finally, we detail our search strategy in Section 3.3. We refer the reader to Figure 1 (b) for a high-level overview of FALCON.

### 3.1 DESIGN SPACE AS A GRAPH

**Motivation**. Previous works generally consider each design choice as isolated from other designs. However, it is often observed that some designs that share the same design features, *e.g.,* graph neural networks (GNNs) that are more than 3 layers and have batch normalization layers, may have similar performances. Moreover, the inductive bias of the relations between design choices can provide valuable information for navigating the design space for the best design. For example, suppose we find that setting batch normalization of a 3-layer GCN (Kipf & Welling, 2017) and a 4-layer GIN (Xu

et al., 2019) to false both degrade the performance. Then we can reasonably infer that a 3-layer GraphSAGE (Hamilton et al., 2017) with batch normalization outperforms the one without. We could leverage such knowledge and only search for the designs that are more likely to improve the task performance. To the best of our knowledge, FALCON is the first method to explicitly consider such relational information among model designs.

**Design graph**. We denote the design graph as $\mathcal{G}(\mathcal{N}, \mathcal{E})$, where the nodes $\mathcal{N}$ include the candidate designs, and edges $\mathcal{E}$ denote the similarities between the candidate designs. Specifically, we use the notion of design distance to decide the graph connectivity, and we elaborate on them below.

**Design distance**. For each numerical design dimension, two design choices have a distance 1 if they are adjacent in the ordered list of design choices. For example, if the hidden dimension size can take values $[16, 32, 64, 128]$, then the distance between 16 and 32 is 1, and the distance between 32 and 128 is 2. For each categorical design dimension, any two distinct design choices have a distance 1. We then define the connectivity of the design graph in terms of the design distance:

**Definition 1 (Design Graph Connectivity)** *The design graph can be expressed as $\mathcal{G}(\mathcal{N}, \mathcal{E})$, where the nodes $\mathcal{N} = \{d_1, \dots, d_n\}$ are model designs, and $(d_i, d_j) \in \mathcal{E}$ iff the design distance between $d_i$ and $d_j$ is 1.*

**Structure of the design graph**. The definition of edges implies that the design graph is highly structured, with the following properties: (1) All designs that are the same except for one categorical design dimension form a clique subgraph. (2) All designs that are the same except $k$ numerical design dimensions form a grid graph structure. Moreover, we use a special calculation for the design distance with a combination of design dimensions that have dependencies. For example, the design dimensions of pooling operations, pooling layers, and the number of layers can depend on each other, thus the design graph structure becomes more complex. See the details in Appendix A.2.

**Design subgraph**. The design graph may contain millions of nodes. Therefore, directly applying the meta-model to the design graph is computationally intensive. Moreover, a reliable performance estimation for an unknown node depends on its similarity between the nodes already explored by the search algorithm. Therefore, we focus on using a meta-model to predict performance for a dynamic subgraph, *i.e.,* design subgraph, containing the explored nodes in the current search stage and the candidate nodes to be sampled in the next step. The candidate set can be constructed by selecting the multi-hop neighbors of explored nodes on the design graph. The design subgraph is defined as:

**Definition 2 (Design Subgraph)** *During a search, suppose the explored node set is $\mathcal{N}_e$ and the candidate set is $\mathcal{N}_c$. The design subgraph is formulated as $\mathcal{G}_s(\mathcal{N}_s, \mathcal{E}_s)$, where $\mathcal{N}_s = \mathcal{N}_e \cup \mathcal{N}_c$ are the nodes and $\mathcal{E}_s = \{(u, v) | u \in \mathcal{N}_s, v \in \mathcal{N}_s, (u, v) \in \mathcal{N}\}$ are the edges.*

Given the design subgraph, we formulate the AutoML problem as searching for the node, *i.e.,* design choice, with the best task performance.

## 3.2 META-GNN FOR PERFORMANCE PREDICTION

Here we introduce a meta-model, named meta-GNN, to predict the performance of model designs, *i.e.,* nodes of the design subgraph. The goal of meta-GNN is learning the inductive bias of design relations, which is used to navigate the search path on the design graph. As is illustrated in Figure 2, the meta-GNN comprises a task-agnostic module and a task-specific module, used to capture the knowledge of model design and task performance, respectively.

**Task-agnostic module**. The task-agnostic module uses a design encoder to encode the design features on nodes of the design subgraph, and a relation encoder to capture the design similarities and differences on edges of the design subgraph. After that, it performs message passing on the design subgraph. We introduce each component below:

- *Design encoder*: it computes the node features of design subgraph by the concatenation of the feature encoding of each design dimension. For numerical design dimensions,we conduct min-max normalization on their values as the node features. For categorical design dimensions such as aggregation operator which takes one of (SUM, MAX, MEAN), we encode it as a one-hot feature.
- *Relation encoder*: it captures the similarity relationships between the connecting designs. For each $(d_i, d_j) \in \mathcal{E}$, we encode the design dimension where $d_i$ and $d_j$ differ by a one-hot encoding.

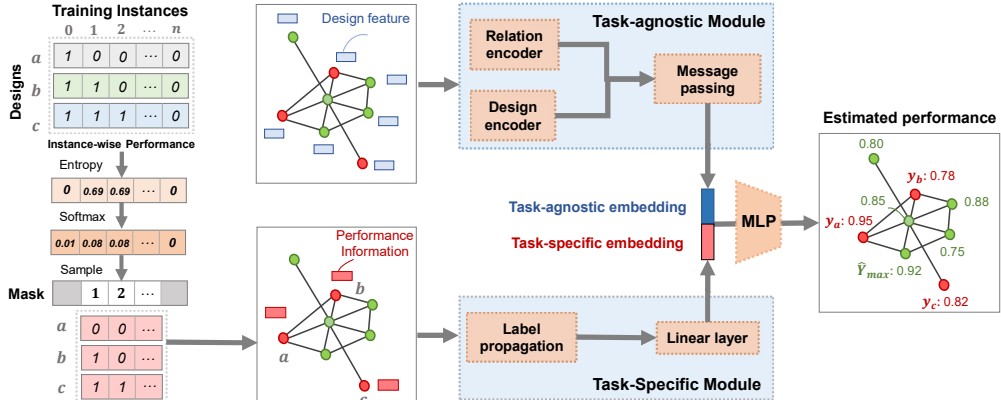

Figure 2: **Meta-GNN Framework**: Task-agnostic module generates the embedding given the design variables and their graphical structures. Task-specific module leverages performance information and conducts label propagation to generate the task-specific embeddings. The two embeddings are concatenated and input into an MLP for predicting the design performance.

- *Message passing module*: a GNN model is used to take the design subgraph and the processed features to perform message passing and output node representations. This information will be combined with the task-specific module to predict the design's performance.

**Task-specific module**. The task-specific module takes into account the information of design performance on selected training instances and thus is specific for one dataset.

The challenge of including such task-specific performance is that it is only available on a very limited set of explored nodes. To overcome the challenge, we use label propagation to propagate the performance information of explored nodes to the unexplored nodes. This is based on our observation that models trained with similar designs typically make similar predictions on instances. We provide an example in Figure 2 to illustrate the task-specific module.

- *Identifying critical instances*: The first step is to identify critical training instances that result in different performances across different designs. Here we use a set of explored designs (anchors) to provide the instance-wise performances. Specifically, the $(i, j)$ element of the top left matrix of Figure 2 represents whether the $i$-th design can correctly predict the label of $j$-th instance. Then, we compute the entropy of each training instance's performance over the anchors. Then we obtain the instance-wise probability via Softmax on the entropy vector, from which we sample instances that result in high variation across designs. The high variation implies that these instances can distinguish good designs from bad ones in the design subgraph, which are informative.

- *Label propagation and linear projection*: Based on the inductive bias of smoothness, we perform label propagation to make the task-specific information available to all candidate designs. Concretely, label propagation can be written as

$$\mathbf{Y}^{(k+1)} = \alpha \cdot D^{-1/2} A D^{-1/2} \mathbf{Y}^{(k)} + (1 - \alpha) \mathbf{Y}^{(k)} \tag{1}$$

where each row of $\mathbf{Y}$ is the performance vector of design $i$ (if explored) or a zero vector (for the unexplored designs). $D \in \mathbb{R}^{|\mathcal{N}_s| \times |\mathcal{N}_s|}$ is the diagonal matrix of node degree, $A \in \mathbb{R}^{|\mathcal{N}_s| \times |\mathcal{N}_s|}$ is the adjacent matrix of the design subgraph, and $\alpha$ is a hyper-parameter. After label propagation, we use a linear layer to project the performance information to another high-dimensional space.

Finally, as shown in Figure 2, we concatenate the output embeddings of the task-specific and task-agnostic modules and use an MLP to produce the performance predictions.

**Objective for Meta-GNN**. The training of neural performance predictor is commonly formulated as a regression using mean square error (MSE), in order to predict how good the candidate designs are for the current task. However, the number of explored designs is usually small for sample-efficient AutoML, especially during the early stage of the search process. Thus, the deep predictor tends to overfit, degrading the reliability of performance prediction. To solve this problem, we incorporate a pair-wise rank loss (Burges et al., 2005; Hu et al., 2021) with the MSE objective, resulting in

---

**Algorithm 1:** Pseudocode of FALCON

---

**Require:** $S$: Design space. $K$: Exploration size. $h_\theta$: Meta-GNN. $V/V'$: Warm-up / Full epoch. $\eta$: Learning rate. $C$: Number of start nodes.

1: $\Omega \leftarrow$ SAMPLE-NODES$(S, C)$                  // Initialize the exploration set
2: $\Gamma \leftarrow$ MULTI-HOP-NEIGHBORS$(\Omega)$       // Construct candidate set from multi-hop neighbors
3: $Y_\Omega =$ GET-VALIDATION-PERFORMANCE$(\Omega, V)$     // Explore the initial nodes (for $V$ epochs)
4: **while** $t = |\Omega| < K$ **do**
5:      $\mathcal{G}_s^{(t)} \leftarrow (\mathcal{N}_v = \Omega \cup \Gamma, \mathcal{E} =$ SIMILARITY$(\mathcal{N}_v)$         // (1) Update the design subgraph
6:      **while** not converge **do**
7:          $\theta \leftarrow \theta - \eta \cdot \partial\mathcal{L}(\hat{Y}_\Omega, h_\theta(\mathcal{G}_s^{(t)})_\Omega)/\partial\theta$      // (2) Compute Eq. 2 and conduct optimization
8:      **end while**
9:      // (3) Sample a candidate node with probability proportional to the meta-GNN's prediction
10:      $d^{(t)} =$ SAMPLE-WITH-PROBABILITY$(\Gamma,$ Softmax$(h_\theta(\mathcal{G}_s^{(t)})_\Gamma))$
11:      $Y_t =$ GET-VALIDATION-PERFORMANCE$(d^{(t)}, V)$    // (2) Explore the current selected node
12:      $\Omega \leftarrow \Omega \cup \{d^{(t)}\}, \Gamma \leftarrow \Gamma \cup$ MULTI-HOP-NEIGHBORS$(\{d^{(t)}\})$
13: **end while**
14: $D =$ SELECT-TOPK$(\{\Omega_i : Y_i\}_{i=1}^K,$ size $=$ MIN$(\lceil 10\% \cdot K \rceil, 5))$    // Models to be fully trained
15: $Y' =$ GET-VALIDATION-PERFORMANCE$(D, V')$           // Obtain the final performance
16: $I =$ ARGMAX$(Y')$                                    // Obtain best model
17: **return** $D_I, Y'_I$

---

a quadratic number of training pairs, thus reducing overfitting. Furthermore, predicting relative performance is more robust across datasets than predicting absolute performance. Overall, the objective is formulated as follows:

$$\mathcal{L}(\hat{Y}, Y) = \sum_{i=1}^{N}(\hat{Y}_i - Y_i)^2 + \lambda\mathcal{L}_{rank}(\hat{Y}, Y), \text{ where}$$

$$\mathcal{L}_{rank}(\hat{Y}, Y) = \sum_{i=1}^{N}\sum_{j=i}^{N}(-1)^{\mathbb{I}(Y_i > Y_j)} \cdot \sigma\left(\frac{\hat{Y}_i - \hat{Y}_j}{\tau}\right)$$

(2)

where $\lambda$ is the trade-off hyper-parameter, $\tau$ is the temperature controlling the minimal performance gap that will be highly penalized, and $\sigma$ is the Sigmoid function. Thus, the meta-GNN is trained to predict the node performance on the design subgraph supervised by the explored node performance.

### 3.3 SEARCH STRATEGY

Equipped with the meta-GNN, we propose a sequential search strategy to search for the best design in the design graph. The core idea is to leverage meta-GNN to perform fast inference on the dynamic design subgraph, and decide what would be the next node to explore, thus navigating the search. We summarize our search strategy in Algorithm 1. Concretely, our search strategy consists of the following three steps:

- *Initialization*: As shown in Figure 1 (b), FALCON randomly samples multiple nodes on the design graph. The motivation of sampling multiple nodes in the initial step is to enlarge the receptive field on the design graph to avoid the local optima and bad performance, which is empirically verified in Appendix D. Then, FALCON explore the initialized nodes by training designs on the tasks and construct the instance mask for the task-specific module.

- *Meta-GNN training*: Following Figure 2, meta-GNN predicts the performance of the explored nodes. The loss is computed via Equation 2 and back-propagated to optimize the meta-GNN.

- *Exploration via inference*: Meta-GNN is then used to make predictions for the performances of all candidate nodes. Then we apply Softmax on the predictions as the probability distribution of candidate designs, from which FALCON samples a new node and updates the design subgraph.

At every iteration, FALCON extends the design subgraph through the last two steps. After several iterations, it selects and retrains a few designs in the search trajectory with top performances. Overall,

FALCON approaches the optimal design navigated by the relational inductive bias learned by meta-GNN, as shown in Figure 1 (b).

## 4 EXPERIMENTS

We conduct extensive experiments on 27 graph datasets and an image dataset. The goal is twofold: (1) to show FALCON's sample efficiency over the existing AutoML methods (*cf.* Section 4.2) and (2) to provide insights into how the inductive bias of design relartions navigate the search on design graph (*cf.* Section 4.3).

### 4.1 EXPERIMENTAL SETTINGS

We consider the following tasks in our evaluation and we leave the details including dataset split, evaluation metrics, and hyper-parameters in Appendix A.

**Node classification**. We use 6 benchmarks ranging from citation networks to product or social networks: Cora, CiteSeer, PubMed (Sen et al., 2008), ogbn-arxiv (Hu et al., 2020), AmazonComputers (Shchur et al., 2018), and Reddit (Zeng et al., 2020).

**Graph classification**. We use 21 benchmark binary classification tasks in TUDataset (Morris et al., 2020), which are to predict certain properties for molecule datasets with various distribution.

**Image classification**. We use CIFAR-10 (Krizhevsky, 2009). See details in Appendix C.

**Baselines**. We compare FALCON with three types of baselines:

- Simple search strategies: Random, Simulated Annealing (SA), Bayesian Optimization (BO) (Bergstra et al., 2011).
- AutoML approaches: DARTS (Liu et al., 2019), ENAS (Pham et al., 2018), GraphNAS (Gao et al., 2019), AutoAttend (Guan et al., 2021), GASSO (Qin et al., 2021), where the last three methods are specifically designed for graph tasks.
- **Ablation** models: FALCON-G and FALCON-LP, where FALCON-G discards the design graph and predicts the design performance using an MLP, and FALCON-LP removes the task-specific module and predicts design performance using only the task-agnostic module.

We also include a naive method, BRUTEFORCE, which trains 5% designs from scratch and returns the best design among them. The result of BRUTEFORCE is regarded as the approximated ground truth performance. We compare FALCON and the simple search baselines under *sample size controlled search*, where we limit the number of explored designs. We set the exploration size as 30 by default.

**Design Space**. We use different design spaces on node- and graph-level tasks. Specifically, The design variables include common hyper-parameters, *e.g.,* dropout ratio, and architecture choices, *e.g.,* layer connectivity and batch normalization. Moreover, we consider node pooling choices for the graph classification datasets, which is less studied in the previous works (Cai et al., 2021; Gao et al., 2019; Zhou et al., 2019). Besides, we follow You et al. (2020b) and control the number of parameters for all the candidate designs to ensure a fair comparison. See Appendix A.2 for the details.

### 4.2 MAIN RESULTS

**Node classification tasks**. Table 1 summarizes the performance of FALCON and the baselines.

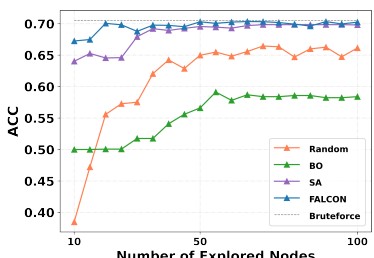

Figure 3: Accuracy *v.s.* the number of explored nodes on ogbn-arxiv.

Notably, FALCON takes comparable search cost as the one-shot methods and is 15x less expensive than GraphNAS. Moreover, FALCON achieves the best performances over the baselines with sufficient margins in the most datasets, using only 30 explored designs. For example, FALCON outperforms ENAS by 1.8% in CiteSeer and GASSO by 1.6% in AmazonComputers. Also, the removal of the design graph and task-specific module decreases the performance constantly, which validates their effectiveness. It is worth mentioning that FALCON is competitive with BRUTEFORCE, demonstrating the excellence of FALCON in searching for globally best-performing designs.

Table 1: Search results on five node classification tasks, where Time stands for the search cost (GPU·hours). We conduct t-test to compute p-value on our method with the best AutoML baselines.

| | Cora | | CiteSeer | | Pubmed | | AmazonComputers | | Reddit | |
|---|---|---|---|---|---|---|---|---|---|---|
| | ACC | Time | ACC | Time | ACC | Time | ACC | Time | F1 | Time |
| Random | $80.8_{\pm1.7}$ | 0.20 | $71.2_{\pm0.8}$ | 0.22 | $86.0_{\pm3.5}$ | 0.24 | $81.6_{\pm3.0}$ | 0.16 | $94.3_{\pm0.1}$ | 0.97 |
| BO | $85.1_{\pm0.3}$ | 0.28 | $72.6_{\pm0.9}$ | 0.30 | $88.5_{\pm0.3}$ | 0.31 | $82.3_{\pm6.3}$ | 0.16 | $94.2_{\pm0.2}$ | 0.94 |
| SA | $81.1_{\pm0.8}$ | 0.24 | $74.7_{\pm0.2}$ | 0.25 | $88.9_{\pm0.1}$ | 0.29 | $81.2_{\pm6.9}$ | 0.23 | $94.3_{\pm0.5}$ | 0.97 |
| ENAS | $85.8_{\pm0.4}$ | 0.27 | $74.9_{\pm0.2}$ | 0.39 | $88.6_{\pm0.8}$ | 2.06 | $74.5_{\pm1.2}$ | 0.83 | $92.3_{\pm1.0}$ | 1.98 |
| DARTS | $85.8_{\pm0.2}$ | 0.25 | $75.2_{\pm0.3}$ | 0.25 | $89.1_{\pm0.1}$ | 0.35 | $84.1_{\pm1.9}$ | 0.35 | [OoM] | - |
| GraphNAS | $82.2_{\pm3.6}$ | 3.12 | $74.9_{\pm0.6}$ | 3.99 | $89.2_{\pm0.3}$ | 5.37 | $88.5_{\pm2.4}$ | 2.53 | $89.1_{\pm2.9}$ | 3.03 |
| AutoAttend | $84.6_{\pm0.2}$ | 1.23 | $73.9_{\pm0.2}$ | 1.25 | $84.4_{\pm0.7}$ | 1.55 | $87.3_{\pm1.1}$ | 2.62 | [OoM] | - |
| GASSO | $\mathbf{86.8_{\pm1.1}}$ | 0.38 | $75.3_{\pm0.7}$ | 0.33 | $86.3_{\pm0.4}$ | 0.41 | $89.8_{\pm0.1}$ | 0.73 | [OoM] | - |
| FALCON-G | $84.5_{\pm0.8}$ | 0.23 | $74.3_{\pm1.7}$ | 0.24 | $89.2_{\pm0.1}$ | 0.26 | $87.6_{\pm0.9}$ | 0.27 | $93.7_{\pm0.4}$ | 1.11 |
| FALCON-LP | $85.5_{\pm1.0}$ | 0.26 | $74.6_{\pm0.1}$ | 0.26 | $89.0_{\pm0.2}$ | 0.29 | $90.7_{\pm0.6}$ | 0.30 | $94.9_{\pm0.2}$ | 1.00 |
| FALCON | $\mathbf{86.4_{\pm0.5}}$ | 0.26 | $\mathbf{76.2_{\pm0.4}}$ | 0.28 | $\mathbf{89.3_{\pm0.5}}$ | 0.32 | $\mathbf{91.2_{\pm0.5}}$ | 0.30 | $\mathbf{95.2_{\pm0.2}}$ | 1.15 |
| BRUTEFORCE | 87.0 | 52.5 | 76.0 | 59.7 | 90.0 | 63.0 | 91.4 | 81.5 | 95.5 | >200 |
| p-value | - | - | 0.051 | - | 0.145 | - | 0.017 | - | 0002 | - |

We further investigate the speed-performance trade-off of FALCON and other sample-based approaches in ogbn-arxiv. We run several search trials under different sample sizes. As shown in Figure 3, FALCON reaches the approximated ground truth result with very few explored nodes. In contrast, SA and Random require more samples to converge, while BO performs bad even with a large number of explored nodes, potentially due to its inability in dealing with high-dimensional design features.

**Graph classification tasks**. The graph classification datasets cover a wide range of graph distributions. In Table 2, we report the selected performance results for graph classification tasks and leave other results including the search costs in Appendix B. We highlight three observations:

- On average, the state-of-the-art AutoML baselines algorithms perform close to the simple search methods, indicating the potentially unreliable search, as similarly concluded by Yu et al. (2020).

- FALCON surpasses the best AutoML baselines with an average improvement of 3.3%. The sufficient and consistent improvement greatly validates our sample efficiency under a controlled sample size. where FALCON can explore the designs that are more likely to perform well through the relational inference based on the relations of previously explored designs and their performances.

- In the second block, we attribute the high sample efficiency of FALCON to the exhibition of design relations and the performance information from the training instances. Specifically, FALCON outperforms FALCON-LP by 4.87% on average, indicating that the task-specific module provides more task information that aids the representation learning of model designs, enabling a fast adaption on a certain task. Moreover, FALCON gains an average improvement of 6.43% compared to FALCON-G, which justifies our motivation that the design relations promote the learning of relational inductive bias and guide the search on the design graph.

We also conduct experiments similar to Figure 3 to investigate how FALCON converges with the increasing sample size (*cf.* Appendix B.1) and report the best designs found by FALCON for each dataset (*cf.* Appendix B.2). Besides, we provide sensitivity analysis on FALCON's hyper-parameters, *e.g.,* number of random start nodes $C$ (*cf.* Appendix D).

**Image classification task**. We demonstrate the potential of FALCON in image domain. Due to space limitation, we leave the results of CIFAR-10 to Appendix C. We found FALCON can search for designs that are best-performing, compared with the baselines. Specifically, it gains average improvements of 1.4% over the simple search baselines and 0.3% over the one-shot baselines on the architecture design space, with search cost comparable to the one-shot based baselines.

## 4.3 CASE STUDIES OF FALCON

We study FALCON in two dimensions: (1) Search process: we probe FALCON's inference process through the explanations of meta-GNN on a design graph, and (2) Design representations: we visualize the node representations output by the meta-GNN to examine the effect of design choices.

**Search process**. We use GNNExplainer (Ying et al., 2019) to explain the node prediction of meta-GNN and shed light on the working mechanism of FALCON. Here we consider the importance of

Table 2: Selected results for the graph classification tasks. The average task performance (ROC-AUC) of the architectures searched by FALCON is 3.3% over the best AutoML baselines.

| | ER-MD | AIDS | OVCAR-8 | MCF-7 | SN12C | NCI109 | Tox21-AhR | Avg. |
|---|---|---|---|---|---|---|---|---|
| Random | $77.5_{\pm1.6}$ | $97.0_{\pm1.4}$ | $56.2_{\pm0.0}$ | $58.2_{\pm0.3}$ | $57.4_{\pm1.0}$ | $73.4_{\pm0.9}$ | $75.7_{\pm2.0}$ | 70.8 |
| BO | $77.6_{\pm3.5}$ | $96.1_{\pm1.0}$ | $63.6_{\pm0.7}$ | $60.7_{\pm0.0}$ | $54.8_{\pm1.1}$ | $73.6_{\pm1.2}$ | $75.5_{\pm1.1}$ | 71.7 |
| SA | $75.9_{\pm4.2}$ | $95.4_{\pm0.9}$ | $59.5_{\pm3.2}$ | $56.7_{\pm0.8}$ | $60.4_{\pm1.7}$ | $76.6_{\pm5.6}$ | $76.5_{\pm3.0}$ | 71.6 |
| ENAS | $76.0_{\pm2.2}$ | $97.1_{\pm0.4}$ | $56.0_{\pm1.3}$ | $59.7_{\pm0.8}$ | $66.4_{\pm0.6}$ | $71.2_{\pm1.0}$ | $73.6_{\pm0.9}$ | 71.4 |
| DARTS | $75.0_{\pm0.7}$ | $\mathbf{98.0_{\pm0.0}}$ | $56.8_{\pm0.3}$ | $60.2_{\pm0.7}$ | $66.0_{\pm0.4}$ | $73.5_{\pm0.2}$ | $76.0_{\pm1.1}$ | 72.2 |
| GraphNAS | $76.9_{\pm3.6}$ | $95.9_{\pm0.8}$ | $58.7_{\pm0.8}$ | $61.3_{\pm5.2}$ | $60.7_{\pm1.5}$ | $73.6_{\pm2.9}$ | $70.6_{\pm4.3}$ | 71.1 |
| AutoAttend | $73.1_{\pm0.8}$ | $97.4_{\pm0.3}$ | $59.8_{\pm0.8}$ | $64.4_{\pm0.2}$ | $71.8_{\pm0.3}$ | $75.9_{\pm1.8}$ | $74.1_{\pm0.9}$ | 73.8 |
| GASSO | $73.2_{\pm0.4}$ | $95.2_{\pm0.7}$ | $62.3_{\pm0.3}$ | $62.5_{\pm0.4}$ | $70.9_{\pm2.3}$ | $73.9_{\pm0.4}$ | $70.2_{\pm3.5}$ | 72.6 |
| FALCON-G | $78.3_{\pm3.0}$ | $96.3_{\pm1.4}$ | $56.4_{\pm1.1}$ | $62.3_{\pm4.5}$ | $69.8_{\pm2.2}$ | $70.3_{\pm6.4}$ | $72.5_{\pm2.8}$ | 72.3 |
| FALCON-LP | $76.7_{\pm2.4}$ | $96.0_{\pm0.2}$ | $61.5_{\pm4.9}$ | $59.5_{\pm5.7}$ | $70.3_{\pm3.8}$ | $73.1_{\pm0.3}$ | $76.5_{\pm2.5}$ | 73.3 |
| FALCON | $\mathbf{78.4_{\pm0.2}}$ | $97.5_{\pm1.1}$ | $\mathbf{66.7_{\pm3.4}}$ | $\mathbf{65.5_{\pm2.5}}$ | $\mathbf{73.3_{\pm0.0}}$ | $\mathbf{78.4_{\pm2.3}}$ | $\mathbf{78.5_{\pm1.1}}$ | $\mathbf{76.9}$ |
| BRUTEFORCE | 83.3 | 96.0 | 67.4 | 70.6 | 73.7 | 81.8 | 82.0 | 79.3 |
| p-value | 0.155 | - | 0.008 | 0.035 | <0.001 | 0.096 | 0.018 | - |

each design dimension for each node's prediction. We demonstrate on a real case when searching on CIFAR-10 (*cf.* Table 12 for the design space). For conciseness, we focus on two design dimensions: (Weight Decay, Batch Size). Then, given a node of interest $n' = (0.9, 128)$, we observe the change in its predictions and dimension importance during the search process.

| Explored node $n_t$: | ... | (0.99, 64) | (0.9, 64) | ... | (0.99, 128) | ... |
|---|---|---|---|---|---|---|
| Performance of $n_t$: | ... | ++ | − | ... | + | ... |
| Prediction on $n'$: | ... | 0.90 | 0.77 | ... | 0.89 | ... |
| Dimension importance: | ... | [0.5, 0.5] | [0.8, 0.2] | ... | [0.6, 0.4] | ... |

Where $+$ and $-$ indicate the relative performance of the explored nodes, $t$ is the current search step. Interestingly, we see that the prediction on $n'$ and the dimension importance evolve with the explored designs and their relations. For example, when weight decay changes from $0.99$ to $0.9$, there is a huge drop in the node performance, which affects the prediction of $n'$ and increases the importance of Weight Decay as design performance seems to be sensitive to this dimension.

**Design representations**. In Figure 4, we visualize the high-dimensional design representations via T-SNE (van der Maaten & Hinton, 2008) after training the meta-GNN on the Cora dataset.

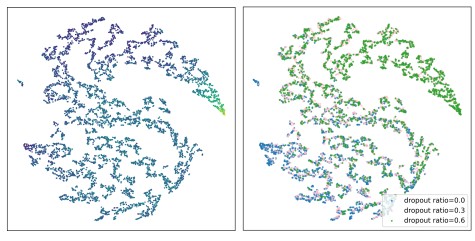

Figure 4: T-SNE visualization for the design representations on Cora dataset.

In the left figure, the better the design performance, the darker the color. Generally, the points with small distance have similar colors or performances, indicating that meta-GNN can distinguish "good" nodes from "bad" nodes. For the right figure, different colors represent different dropout ratios. The high discrimination indicates that the dropout ratio is an influential variable for learning the design representation, which further affects design performance. This evidence validates the meta-GNN's expressiveness and capacity to learn the relational inductive bias between the design choices.

## 5 CONCLUSION, LIMITATION, AND FUTURE WORK

This work introduces FALCON, an efficient sample-based AutoML framework. We propose the concept of design graph that explicitly models and encodes relational information among model designs. On top of the design graph, we develop a sample-efficient strategy to navigate the search on the design graph with a novel meta-model. One future direction is to better tackle the high average node degree on the design graphwhich could cause over-smoothing, especially when the design variables include many categorical variables. And a simple solution is to use edge dropout to randomly remove a portion of edges at each training epoch. Another future direction is to better adapt FALCON on continuous design variables via developing a dynamic design graph that enable a more fine-grained search between the discretized values.

## REPRODUCIBILITY STATEMENT

All of the datasets used in this work are public. For experimental setup, we state the detailed settings in Appendix A and Appendix C, including the graph pre-processing, dataset splits, hyper-parameters. Moreover, we include our code in an anonymous link for public access. For the results, we report the best models found by our algorithm as well as their corresponding performances. Overall, we believe we have made great efforts to ensure reproducibility in this paper.

## ETHICS STATEMENT

In this work, we propose a novel algorithm to search for the best model designs where no human subject is related. This work could promote the discovery of more powerful and expressive models and provide insights into design relations. However, while best-performing models may be "experts" in fulfilling given tasks, they are not necessarily fair towards different user or entity groups. We believe this is a general issue in the AutoML area and should be well addressed to ensure the ethics of models in real-world applications.

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

# A EXPERIMENT DETAILS

## A.1 SETTINGS

**Graph classification datasets**. The graph classification datasets used in this work are summarized in Table 3. And the detailed dataset statics can be referred from https://chrsmrrs.github.io/datasets/docs/datasets/.

Table 3: List of the graph classification datasets used in this work.

| Small Scale | AIDS, BZR-MD, COX2-MD, DHFR-MD, Mutagenicity, NCI1, NCI109, PTC-MM, PTC-MR |
|---|---|
| Medium/Large Scale | Tox21-AhR, MCF-7, MOLT-4, UACC257, Yeast, NCI-H23, OVCAR-8, P388, PC-3, SF-295, SN12C, SW-620 |

Specifically, all datasets are binary classification tasks that predict certain properties for small molecules. For example, the labels in Tox21-AhR represent toxicity/non-toxicity, while the graphs in Mutagenicity are classified into two classes based on their mutagenic effect on a bacterium (Morris et al., 2020). Consequently, we use atom types as the node features and bond types as edge features.

**Evaluation metrics**. For Reddit, we use F1 score (micro) as the evaluation metric following the previous work (Zeng et al., 2020). For other node classification tasks and image dataset, we use classification accuracy as the evaluation metric. For the graph classification tasks, we use ROC-AUC as the evaluation metric.

**Dataset splits**. For ogbn-arxiv and Reddit, we use the standardized dataset split. For other node classification datasets, we split the nodes in each graph into 70%, 10%, 20% in training, validation, and test sets, respectively. For graph classification tasks, we split the graphs into 80%, 10%, 10% for training, validation, and test sets, respectively.

**Hyper-Parameters**. We tuned the hyper-parameters of the baselines based on the default setting in their public codes. For FALCON, we construct the candidate set as the 3-hop neighbors of the explored nodes and set the number of start nodes as $\min(\lceil 10\% \cdot K \rceil, 10)$, where $K$ denotes the exploration size. The meta-GNN is constitute of 3 message-passing layers and 3 label propagation layers. All the experiments are repeated at least 3 times.

## A.2 DESIGN SPACES

### A.2.1 DESIGN SPACES FOR THE SAMPLE-BASED METHODS

In this work, we use different design spaces for the datasets depending on the task types, *i.e.,* node or graph level. We summarize the design variables and choices in Table 4 and Table 5. For the design space of Reddit, we replace "Aggregation" in Table 4 with "Convolutional layer type", which takes values from {GCNConv (Kipf & Welling, 2017), SAGEConv (Hamilton et al., 2017), GraphConv (Morris et al., 2019), GINConv (Xu et al., 2019), ARMAConv (Bianchi et al., 2019), TAGConv (Du et al.)}.

Table 4: Design Space for the node-level tasks (except for Reddit). 5,832 candidates in total.

| Type | Variable | Candidate Values |
|---|---|---|
| Hyper-parameters | Dropout ratio | [0.0, 0.3, 0.6] |
| Architecture | # Pre-process layers
# Message passing layers
# Post-precess layers
Layer connectivity
Activation
Batch norm
Aggregation | [1, 2, 3]
[2, 4, 6, 8]
[1, 2, 3]
STACK, SUM, CAT
ReLU, Swish, Prelu
True, False
Mean, Max, SUM |

Table 5: Design Space for the graph-level tasks. 58,320 candidates in total.

| Type | Variable | Candidate Values |
|---|---|---|
| Hyper-parameters | Dropout ratio | [0.0, 0.3, 0.6] |
| Architecture | # Pre-process layers
# Message passing layers
# Post-precess layers
Layer connectivity
Activation
Batch norm
Aggregation
Node pooling flag (Use node pooling)
Node pooling type (if applicable)
Node pooling loop | [1, 2, 3]
[2, 4, 6, 8]
[1, 2, 3]
STACK, SUM, CAT
ReLU, Swish, Prelu
True, False
Mean, Max, SUM
True, False
TopkPool (Gao & Ji, 2019), SAGPool (Lee et al., 2019),
PANPool (Ma et al., 2020), EdgePool (Diehl, 2019)
[2, 4, 6] |

Specifically, The STACK design choice means directly stacking multiple GNN layers, *i.e.,* without skip-connections. We also support node pooling operations for graph classification tasks, where the pooling loop stands for the number of message passing layers between each pooling operation. If the number of message passing layers is $m$ and the node pooling loop is $l$, there will be a node pooling layer after the $i$th message passing layer in the design model (hierarchical pooling), where $i \in \{1 + k \cdot l \mid k = 0, \ldots, \lceil (m-1)/l \rceil - 1\}$. Moreover, to avoid duplicated and invalid designs, some design variables are required to satisfy specific dependency rules, and we take two examples to elaborate on this point.

- *If the node pooling flag of a design is False, then the design does not have any value on node pooling type and node pooling loop, and vice versa.*

  For example, we denote node pooling flag as $f$, node pooling type as $t$, and $*$ as any design choice, then($f$=False, $t$=$*$) or ($f$=False, $l$=*) will both be invalid.

- *The node pooling loop should not exceed the number of message passing layers.*

  For example, design $A$ ($m$=4, $l$=4) and design $B$ ($m$=4, $l$=6) that take the same values on other design variables are duplicated.

Thus, the design graph constructed under dependency rules is more complex. Without loss of generality, we define that the distance of ($f$=False) and ($f$=True, $l$=MIN($\{i \in \mathbb{L}\}$)) as 1, where $\mathbb{L}$ represents the design choice of node pooling loop. Thus, the design graph is a connected graph that enables the exploration of any node with random initialization. It is also worth mentioning that the search strategy of FALCON is modularized given the design graph. In contrast, the dependency rules constrain the action space of reinforcement learning methods, *e.g.,* ($f$=Ture $\rightarrow$ False) is inapplicable, which requires special operation inside the controller.

Table 7: Design space for the one-shot baselines on node and graph classification tasks.

| Variable | Candidate Values |
|---|---|
| Dropout ratio | [0.0, 0.3, 0.6] |
| Layer connectivity | STACK, SUM |
| # Pre-process layers | [1, 2, 3] |
| # Message passing layers | [2, 4, 6, 8] |
| # Post-precess layers | [1, 2, 3] |
| Activation | ReLU, Swish, Prelu |
| Batch norm | True, False |
| Aggregation | Mean, Max, SUM |

Table 6: Statistics and the construction time of the design graphs.

| | #Nodes | #Edges (Undirected) | Ave. Degree | Diameter | construction time (s) |
|---|---|---|---|---|---|
| DG-1 | 5,832 | 78.732 | 13.5 | 13 | 13 |
| DG-2 | 58,320 | 1,070,172 | 18.4 | 17 | 147 |

We further summarize the statistics and construction time of the design graphs in Table 6, where DG-1 and DG-2 denote the design graphs for node-level and graph-level tasks, respectively. We use multi-processing programing on 50 CPUs (Intel Xeon Gold 5118 CPU @ 2.30GHz) to conduct the graph construction. Note that we don't have to construct the entire design graph in the pre-processing step, since we only extend the small portion of the design graph, *i.e.,* , the design subgraph, during the search process. Thus, the total time cost of constructing the design subgraph will be $O(E')$ where $E'$ is the number of edges in the design subgraph, which largely lowers the time costs.

### A.2.2 DESIGN SPACES FOR THE ONE-SHOT BASELINES

The one-shot models (Liu et al., 2019; Pham et al., 2018) is built upon a super-model that is required to contain all of the architecture choices. We build the macro search space over entire models for both node classification and graph classification datasets with constraints. Firstly, we do not consider CAT (skip-concatenate) a layer connectivity choice, and we also remove design variables for node pooling. The reason is that CAT and node pooling operations change the input shape and make the output embeddings inapplicable for the subsequent weight-sharing modules in our settings. Secondly, the layer connectivity is customized for each layer following the previous works (Liu et al., 2019; Pham et al., 2018), instead of setting as a global value for every layer. Overall, we summarize the design space in Table 7.

To enable a fair comparison, we fine-tune the hyper-parameters and report the best results of the architectures found by the one-shot methods according to their performance in the validation sets.

## B MORE EXPERIMENTAL RESULTS ON GRAPH TASKS

### B.1 GRAPH CLASSIFICATION TASKS

**Task performance**. Here we provide more results of task performance on the graph classification dataset. We repeat each experiment at least 3 times and report the average performances and the standard errors. The results are summarized in Table 8. The results well demonstrate the preeminence of FALCON in searching for good designs under different data distributions.

**Search cost**. As shown in Figure 5, we report the search cost of Random, DARTS, ENAS, GraphNAS, and FALCON on the selected datasets. The time measurements are conducted on a single NVIDIA GeForce 3070 GPU (24G). We see FALCON has a comparable time cost with Random and DARTS, which empirically proves the efficiency of FALCON.

However, as FALCON still needs to sample designs and train them from scratch (*i.e.,* the search cost of FALCON is bounded by the search cost of Random), the computational cost is relatively high in

Table 8: Test performance (ROC-AUC) on the graph classification datasets.

| | DHFR-MD | COX2-MD | Mutagenicity | MOLT-4 | NCI-H23 | PTC-MR | P388 |
|---|---|---|---|---|---|---|---|
| Random | $59.0_{\pm5.2}$ | $63.2_{\pm1.9}$ | $77.1_{\pm1.9}$ | $58.6_{\pm0.7}$ | $58.4_{\pm2.1}$ | $59.9_{\pm5.7}$ | $63.9_{\pm0.7}$ |
| BO | $55.1_{\pm0.0}$ | $\mathbf{71.6}_{\pm\mathbf{5.5}}$ | $78.3_{\pm0.7}$ | $58.1_{\pm2.0}$ | $63.6_{\pm0.0}$ | $58.8_{\pm7.7}$ | $68.9_{\pm0.0}$ |
| SA | $56.0_{\pm7.1}$ | $67.4_{\pm3.1}$ | $81.1_{\pm0.3}$ | $54.8_{\pm1.1}$ | $56.7_{\pm3.2}$ | $59.4_{\pm6.2}$ | $74.4_{\pm1.0}$ |
| ENAS | $53.5_{\pm3.7}$ | $57.9_{\pm1.7}$ | $75.0_{\pm1.6}$ | $61.5_{\pm0.1}$ | $61.2_{\pm1.1}$ | $59.8_{\pm1.6}$ | $68.3_{\pm1.2}$ |
| DARTS | $55.8_{\pm6.3}$ | $70.4_{\pm3.2}$ | $74.4_{\pm0.7}$ | $61.4_{\pm0.7}$ | $63.5_{\pm1.9}$ | $59.3_{\pm0.5}$ | $70.8_{\pm0.7}$ |
| GraphNAS | $61.6_{\pm4.3}$ | $63.9_{\pm2.5}$ | $80.2_{\pm1.5}$ | $62.1_{\pm1.0}$ | $62.4_{\pm3.9}$ | $58.6_{\pm6.7}$ | $68.2_{\pm2.5}$ |
| AutoAttend | $63.3_{\pm0.9}$ | $68.8_{\pm0.7}$ | $79.6_{\pm0.1}$ | $59.5_{\pm0.2}$ | $61.8_{\pm0.2}$ | $57.8_{\pm0.6}$ | $74.9_{\pm0.5}$ |
| GASSO | $60.9_{\pm2.3}$ | $68.5_{\pm2.0}$ | $75.1_{\pm0.5}$ | $57.4_{\pm0.9}$ | $64.7_{\pm1.5}$ | $51.9_{\pm5.2}$ | $71.3_{\pm1.4}$ |
| FALCON-G | $58.5_{\pm8.8}$ | $67.3_{\pm3.6}$ | $79.8_{\pm1.7}$ | $62.8_{\pm3.3}$ | $62.2_{\pm1.2}$ | $55.6_{\pm6.9}$ | $74.2_{\pm3.7}$ |
| FALCON-LP | $61.4_{\pm1.5}$ | $66.8_{\pm3.6}$ | $80.2_{\pm0.8}$ | $58.5_{\pm8.8}$ | $63.7_{\pm4.1}$ | $55.6_{\pm2.0}$ | $75.1_{\pm1.1}$ |
| FALCON | $\mathbf{63.6}_{\pm\mathbf{7.9}}$ | $67.3_{\pm3.2}$ | $\mathbf{81.1}_{\pm\mathbf{0.5}}$ | $\mathbf{64.4}_{\pm\mathbf{4.0}}$ | $\mathbf{66.6}_{\pm\mathbf{3.3}}$ | $\mathbf{60.0}_{\pm\mathbf{1.4}}$ | $\mathbf{77.0}_{\pm\mathbf{1.4}}$ |

| (cont.) PTC-MM | PC-3 | SF-295 | NCI1 | SW-620 | UACC257 | Yeast | Avg. |
|---|---|---|---|---|---|---|---|
| $52.5_{\pm5.6}$ | $60.4_{\pm0.0}$ | $55.3_{\pm0.5}$ | $77.5_{\pm0.3}$ | $57.5_{\pm2.7}$ | $61.1_{\pm0.5}$ | $53.3_{\pm0.0}$ | $61.3$ |
| $60.1_{\pm1.5}$ | $59.7_{\pm0.1}$ | $60.6_{\pm0.3}$ | $77.3_{\pm0.0}$ | $63.8_{\pm0.9}$ | $60.8_{\pm0.0}$ | $55.0_{\pm0.0}$ | $63.7$ |
| $58.1_{\pm4.6}$ | $69.0_{\pm2.0}$ | $55.3_{\pm0.6}$ | $79.6_{\pm5.3}$ | $58.2_{\pm2.1}$ | $64.2_{\pm0.1}$ | $53.8_{\pm1.2}$ | $63.4$ |
| $52.4_{\pm2.9}$ | $62.2_{\pm1.1}$ | $60.9_{\pm2.2}$ | $77.2_{\pm1.2}$ | $64.8_{\pm2.2}$ | $64.7_{\pm0.5}$ | $\mathbf{63.0}_{\pm\mathbf{1.1}}$ | $63.0$ |
| $52.6_{\pm4.1}$ | $61.6_{\pm0.5}$ | $62.2_{\pm1.1}$ | $66.3_{\pm3.0}$ | $66.0_{\pm0.3}$ | $65.7_{\pm0.2}$ | $61.4_{\pm1.4}$ | $63.7$ |
| $54.8_{\pm3.9}$ | $68.6_{\pm2.6}$ | $65.0_{\pm3.1}$ | $78.1_{\pm3.6}$ | $61.8_{\pm4.4}$ | $61.5_{\pm5.1}$ | $57.2_{\pm1.1}$ | $64.6$ |
| $\mathbf{64.7}_{\pm\mathbf{1.2}}$ | $66.2_{\pm0.3}$ | $64.2_{\pm0.9}$ | $79.3_{\pm1.6}$ | $65.5_{\pm0.4}$ | $57.1_{\pm0.5}$ | $59.6_{\pm0.8}$ | $65.9$ |
| $63.2_{\pm0.7}$ | $66.0_{\pm1.8}$ | $\mathbf{65.8}_{\pm\mathbf{0.4}}$ | $76.6_{\pm0.6}$ | $64.7_{\pm1.1}$ | $62.7_{\pm0.2}$ | $60.1_{\pm0.4}$ | $64.9$ |
| $55.2_{\pm2.4}$ | $65.1_{\pm2.8}$ | $64.3_{\pm2.1}$ | $80.6_{\pm0.8}$ | $62.0_{\pm3.0}$ | $61.0_{\pm3.8}$ | $57.5_{\pm1.5}$ | $64.7$ |
| $56.8_{\pm6.1}$ | $68.1_{\pm0.3}$ | $63.5_{\pm3.4}$ | $80.4_{\pm0.5}$ | $65.2_{\pm1.7}$ | $54.5_{\pm1.1}$ | $56.6_{\pm1.7}$ | $64.7$ |
| $57.1_{\pm0.1}$ | $\mathbf{71.0}_{\pm\mathbf{1.7}}$ | $64.4_{\pm0.4}$ | $\mathbf{80.9}_{\pm\mathbf{0.8}}$ | $\mathbf{66.6}_{\pm\mathbf{2.1}}$ | $\mathbf{66.7}_{\pm\mathbf{2.1}}$ | $58.1_{\pm1.7}$ | $\mathbf{67.5}^{*}$ |

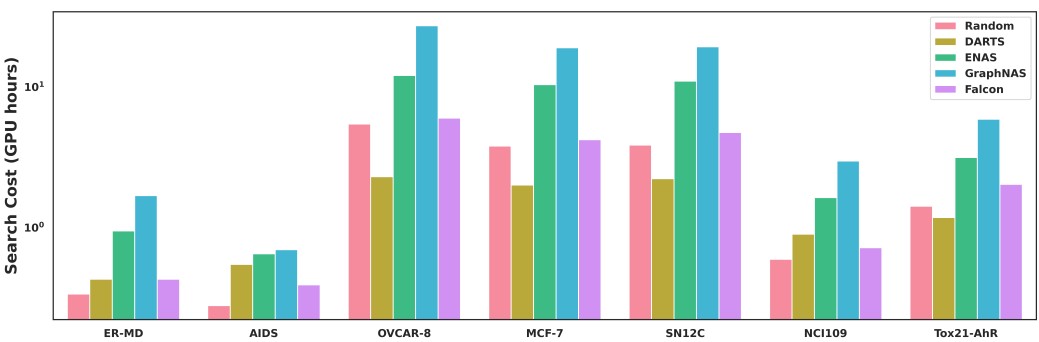

Figure 5: Search cost on the selected datasets.

large datasets, *e.g.,* OVCAR-8 and MCF-7. We can potentially alleviate this limitation via integrating dataset sampling to reduce time costs.

**ROC-AUC *v.s.* exploration size**. Here we report the change in task performance on graph classification datasets with the number of explored nodes. In Figure 6, we visualize the results on two graph classification datasets. We see that FALCON can approach the best-performing designs quickly as the explored size grows.

## B.2 BEST DESIGNS

In Table 9 and Table 10, we summarize the best designs found by FALCON and BRUTEFORCE in each dataset, where the average number of parameters is 137.5k for all the graph classification datasets. Note that we select the best designs according to their performance on validation sets; thus, there are cases where FALCON surpasses BRUTEFORCE. We highlight the design variables that are different between FALCON and BRUTEFORCE for comparison.

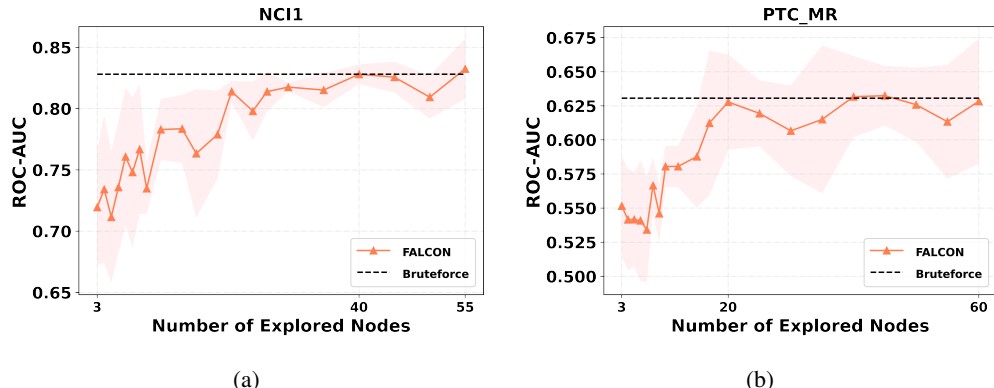

(a)                                                             (b)

Figure 6: Accuracy *v.s.* the number of explored nodes on two graph classification datasets.

Table 9: Average parameters & Best designs in the node classification datasets.

| Dataset | Average Param (k) | | Best design | Test performance (%) |
|---|---|---|---|---|
| ogbn-arxiv | 44.5 | FALCON | *(0.0, 1, 4, 2, STACK, Swish, True, Mean)* | 70.36 |
| | | BRUTEFORCE | *(0.3, 1, 4, 2, SUM, Relu, True, Mean)* | 70.51 |
| Cora | 77.8 | FALCON | *(0.0, 1, 6, 1, SUM, Swish, False, Mean)* | 87.18 |
| | | BRUTEFORCE | *(0.0, 1, 4, 1, STACK, Swish, False, Mean)* | 86.99 |
| Citeseer | 289.0 | FALCON | *(0.3, 1, 2, 1, SUM, Prelu, True, Mean)* | 76.19 |
| | | BRUTEFORCE | *(0.3, 1, 2, 2, SUM, Prelu, False, Mean)* | 75.99 |
| Pubmed | 57.8 | FALCON | *(0.3, 1, 8, 1, SUM, Relu, True, Add)* | 90.04 |
| | | BRUTEFORCE | *(0.3, 1, 8, 1, SUM, Relu, True, Add)* | 90.04 |
| AmazonComputers | 46.2 | FALCON | *(0.0, 1, 4, 1, STACK, Swish, True, Mean)* | 91.64 |
| | | BRUTEFORCE | *(0.0, 3, 4, 2, STACK, Prelu, True, Mean)* | 91.35 |
| Reddit | 47.2 | FALCON | *(0.0, 3, 4, 2, STACK, Prelu, True, ARMAConv)* | 95.46 |
| | | BRUTEFORCE | *(0.0, 3, 4, 2, STACK, Prelu, True, ARMAConv)* | 95.46 |

## B.3 BRUTEFORCE'S CONFIDENCE INTERVAL AND VARIANT

To estimate the uncertainty of Bruteforce, we compute the 95% confidence interval of Bruteforce using bootstrapping. Moreover, we consider a variant of Bruteforce baseline to compare with Bruteforce. Specifically, we train all the designs in the design space for 30 epochs, select the top 10% design, and resume the training until 50 epochs. After that, we choose the top 50% designs to be fully trained and return the best fully trained design based on the validation performance. We run Bruteforce-bootstrap on four datasets as demonstrations. We summarize the results in Table 11.

Table 11: Test performances of Bruteforce and its variant.

| | Cora | CiteSeer | ER-MD | AIDS |
|---|---|---|---|---|
| Bruteforce (max) | 87.0 | 76.0 | 83.3 | 96.0 |
| Confidence interval length | 0.2 | 0.1 | 0.6 | 0.0 |
| Bruteforce-bootstrap | 87.0 | 76.4 | 83.8 | 95.7 |

Surprisingly, we found that the performance of Bruteforce and Bruteforce-bootstrap are very close. This indicates that Bruteforce (fully trained 5% design) is a good surrogate of Bruteforce-bootstrapping (fully trained 5% design, but using bootstrapping selection), and could also well approximate the ground truth performance of the best design.

Table 10: Best designs in the graph classification datasets.

| Dataset | | Best design | Test performance (%) |
|---|---|---|---|
| AIDS | FALCON | *(0.3, 1, 4, 2, SUM, Prelu, True, Add, NoPool)* | 99.02 |
| | BRUTEFORCE | *(0.0, 3, 8, 2, STACK, Relu, True, Add, SAGPool, 2)* | 95.97 |
| COX2-MD | FALCON | *(0.0, 2, 6, 1, SUM, Swish, True, Add, EdgePool, 2)* | 69.87 |
| | BRUTEFORCE | *(0.0, 1, 4, 3, STACK, Prelu, True, Max, TopkPool, 2)* | 65.39 |
| DHFR-MD | FALCON | *(0.3, 2, 4, 2, SUM, Swish, True, Mean, PANPool, 2)* | 71.33 |
| | BRUTEFORCE | *(0.0, 1, 4, 3, CAT, Prelu, True, Mean, SAGPool, 4)* | 56.22 |
| ER-MD | FALCON | *(0.3, 3, 4, 2, CAT, Relu, True, Add, PANPool, 2)* | 81.67 |
| | BRUTEFORCE | *(0.0, 3, 5, 3, CAT, Prelu, True, Max, PANPool, 6)* | 83.33 |
| MCF-7 | FALCON | *(0.6, 1, 8, 2, CAT, Swish, True, Add, NoPool)* | 67.85 |
| | BRUTEFORCE | *(0.0, 1, 2, 6, SUM, Prelu, True, Add, NoPool)* | 70.61 |
| MOLT-4 | FALCON | *(0.0, 3, 6, 2, SUM, Prelu, True, Max, NoPool)* | 69.03 |
| | BRUTEFORCE | *(0.0, 1, 6, 1, SUM, Prelu, True, Add, NoPool)* | 70.30 |
| Mutagenicity | FALCON | *(0.0, 2, 6, 1, CAT, Relu, True, Add, NoPool)* | 81.73 |
| | BRUTEFORCE | *(0.0, 2, 4, 2, CAT, Relu, True, Add, PANPool, 2)* | 81.17 |
| NCI1 | FALCON | *(0.0, 2, 4, 2, SUM, Swish, True, Max, EdgePool, 2)* | 80.13 |
| | BRUTEFORCE | *(0.0, 1, 4, 2, STACK, Prelu, True, Add, EdgePool, 4)* | 82.81 |
| NCI109 | FALCON | *(0.0, 2, 8, 2, STACK, Prelu, True, Max, NoPool)* | 79.98 |
| | BRUTEFORCE | *(0.0, 3, 6, 2, STACK, Prelu, True, Add, EdgePool, 6)* | 81.77 |
| NCI-H23 | FALCON | *(0.0, 2, 6, 1, CAT, Swish, True, Add, NoPool)* | 71.14 |
| | BRUTEFORCE | *(0.0, 2, 6, 1, CAT, Swish, True, Add, NoPool)* | 71.14 |
| OVCAR-8 | FALCON | *(0.6, 2, 6, 2, CAT, Swish, True, Add, NoPool)* | 68.21 |
| | BRUTEFORCE | *(0.6, 1, 2, 3, CAT, Swish, True, Add, NoPool)* | 67.40 |
| P388 | FALCON | *(0.0, 2, 6, 2, CAT, Prelu, True, Add, NoPool)* | 78.75 |
| | BRUTEFORCE | *(0.0, 2, 6, 2, CAT, Prelu, True, Add, NoPool)* | 78.75 |
| PC-3 | FALCON | *(0.0, 2, 6, 1, SUM, Relu, True, Max, NoPool)* | 73.28 |
| | BRUTEFORCE | *(0.0, 2, 6, 1, SUM, Relu, True, Max, NoPool)* | 73.28 |
| PTC-MM | FALCON | *(0.0, 2, 4, 2, STACK, Swish, True, Max, EdgePool, 2)* | 56.96 |
| | BRUTEFORCE | *(0.0, 3, 6, 1, SUM, Swish, True, Mean, EdgePool, 2)* | 52.93 |
| PTC-MR | FALCON | *(0.0, 2, 4, 2, SUM, Swish, True, Mean, TopkPool, 2)* | 60.56 |
| | BRUTEFORCE | *(0.3, 1, 6, 2, SUM, Relu, True, Add, NoPool)* | 63.06 |
| SF-295 | FALCON | *(0.6, 2, 6, 2, CAT, Swish, True, Add, NoPool)* | 64.75 |
| | BRUTEFORCE | *(0.0, 3, 8, 3, SUM, Prelu, True, Max, PANPool, 4)* | 66.47 |
| SN12C | FALCON | *(0.0, 1, 8, 1, CAT, Swish, True, Add, NoPool)* | 73.34 |
| | BRUTEFORCE | *(0.0, 1, 8, 3, CAT, Prelu, True, Add, EdgePool, 6)* | 73.73 |
| SW-620 | FALCON | *(0.0, 2, 4, 2, STACK, Prelu, True, Max, NoPool)* | 69.26 |
| | BRUTEFORCE | *(0.0, 2, 4, 2, STACK, Prelu, True, Max, NoPool)* | 69.26 |
| Tox21-AhR | FALCON | *(0.0, 2, 4, 2, SUM, Prelu, True, Add, EdgePool, 2)* | 79.10 |
| | BRUTEFORCE | *(0.0, 3, 6, 2, SUM, Prelu, True, Add, NoPool)* | 82.02 |
| UACC257 | FALCON | *(0.3, 2, 6, 2, CAT, Swish, True, Max, NoPool)* | 67.94 |
| | BRUTEFORCE | *(0.0, 3, 6, 1, SUM, Prelu, True, Max, NoPool)* | 70.24 |
| Yeast | FALCON | *(0.6, 1, 8, 1, CAT, Prelu, True, Add, NoPool)* | 59.60 |
| | BRUTEFORCE | *(0.0, 1, 2, 3, SUM, Swish, True, Add, EdgePool, 2)* | 60.41 |

## C  EXPERIMENTAL RESULTS ON THE IMAGE TASK

### C.1  DATASET PRE-PROCESSING

We use the CIFAR-10 (Krizhevsky, 2009) image dataset to show FALCON can work well on other machine learning domains. This dataset consists of 50,000 training images and 10,000 test images. We randomly crop them to size $32 \times 32$, and conduct random flipping and normalization.

### C.2  DESIGN SPACES

Here we use two different design spaces to demonstrate FALCON's ability in searching for both hyper-parameters and architectures on image dataset.

**Hyper-parameter Design Space**. We consider a broad space of hyper-parameter search, including common hyper-parameters like Batch Size. We train each design using a SGD optimizer, which requires weight decay and momentum as hyper-parameters. We also use a learning rate (LR) scheduler, which reduces the learning rate when validation performance has stopped improving. Specifically, the scheduler will reduce the learning rate by a factor if no improvement is seen for a 'patience' number of epochs. It is also worth mentioning that FALCON is flexible for other sets of hyper-parameter choices determined by the user end.

Table 12: Hyper-parameter design space for image tasks.

| Variable | Candidate Values |
|---|---|
| Momentum (SGD) | [0.5, 0.9, 0.99] |
| Weight decay | [1e-4, 5e-4, 1e-3, 5e-3] |
| Batch size | [32, 64, 128, 256] |
| LR decay factor | [0.1, 0.2, 0.5, 0.8] |
| LR decay patience | [3, 5, 10] |

**Architecture Design Space**. We construct micro design space for the computational cells. Each cell constitutes of two branches, where we enable five selections: separable convolution with kernel size 3 × 3 and 5 × 5, average pooling and max pooling with kernel size 3×3, and identity. In each branch, we have one dropout layer, where the dropout ratio is one of $\{0.0, 0.3, 0.6\}$, and one batch normalization layer. We also use one of identity and skip-sum as skip-connection within each branch. After the input data is separately computed in each branch, the outputs are added as the final cell output, which is different with the original ENAS paper that searches the computational DAG on the defined nodes.

### C.3  EXPERIMENTAL RESULTS

Here we set the exploration size as 20 for all the sample-based methods. For hyper-parameter design space, we compare FALCON with the baselines that are available for hyper-parameter tuning. Moreover, to accelerate the search process, FALCON explores an unknown design by fine-tuning a pretrained model for several epochs based on the selected hyper-parameters, instead of training each candidate design from scratch. The results are summarized in Table 13.

Table 13: Search results on the hyper-parameter design space for CIFAR-10 dataset.

| | Average Error (%) |
|---|---|
| Random | 4.13 |
| BO | 4.95 |
| SA | 4.04 |
| FALCON | 3.80 |

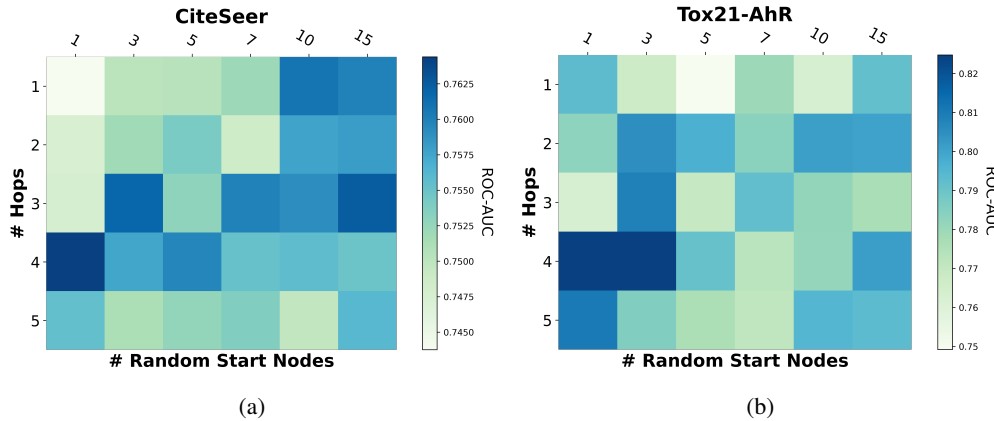

Figure 7: Sensitivity of FALCON's hyper-parameters.

For the architecture design space. For ENAS and DARTS, the learning rate is 0.01, and the maximum training epoch is 300. We repeat each experiment three times and summarize the results in Table 14.

Table 14: Search results on the architecture design space for CIFAR-10 dataset.

|  | Average Error (%) | Search Cost (GPU days) |
| --- | --- | --- |
| Random | 10.43 | 0.64 |
| BO | 9.83 | 0.67 |
| SA | 10.16 | 0.62 |
| ENAS-micro | 9.20 | 0.77 |
| DARTS-micro | 8.97 | 0.89 |
| FALCON | 8.87 | 0.81 |

# D  SENSITIVITY ANALYSIS

We analyze the sensitivity of the hyper-parameters in the search strategy of FALCON, using a node classification task, CiteSeer, and a graph classification task, Tox21-AhR. Specifically, we study the influence of the number of random starting nodes and the candidate scale resulting from an explored node (*i.e.,* how many hop neighbors of an explored node are to be included in the candidate set).

As shown in Figure 7, we find that FALCON outperforms the best AutoML baselines with a large range of hyper-parameters. Specifically, 73% and 97% hyper-parameter combinations of FALCON rank best among the baselines in CiteSeer and Tox21-AhR, respectively.

Moreover, we discover an interesting insight about the size of *receptive field*, *i.e.,* the number of design candidates, during the search process of FALCON. According to the construction of design subgraph, the receptive field size is $O(rh^d)$, where $r$ is the number of random start nodes, $h$ is the number of neighbor hops, and $d$ is the average node degree. We find that the performance of design searched by FALCON increases with the receptive field's size until it reaches a certain scale.

Such patterns have been widely observed in multiple datasets. While the receptive field on the design subgraph should contain sufficient candidates for sampling good ones, it should also prune inferior design space, which doesn't provide insights on navigating the best-performing designs. Thus, the size of the receptive field may be a crucial factor influencing the search quality of FALCON.

