# OpenReview forum: "Efficient Automatic Machine Learning via Design Graphs"
_ICLR.cc/2023/Conference — Submitted to ICLR 2023_

### Official Review · Reviewer_ZhyY · 2022-10-24

**Confidence:** 4
**Correctness:** 2
**Technical Novelty And Significance:** 3
**Empirical Novelty And Significance:** 3
**Recommendation:** 5

**Clarity, Quality, Novelty And Reproducibility:**

The work in this paper seems novel and code has been provided to enable the reader to reproduce the results (although I have not tried to run it). However, the paper is not written to a high standard and many details are unclear.

**Strength And Weaknesses:**

Strengths:
- The proposed approach is novel to the best of my knowledge.
- The experimental results show that FALCON out-performs simple baselines like random search, simulated annealing and Bayesian optimization, as well as a variety of generic AutoML approaches (DARTS, ENAS) as well as AutoML methods designed specifically for graph tasks (AutoAttend, GraphNAS, GASSO).
- The ablation studies show that (a) the search is significantly improved by using the meta-GNN and (b) the meta-GNN is significantly improved by using the task-specific component.

Weaknesses:
- There is some confusing terminology in this paper. It is stated that FALCON "learns the inductive biases" that are present in the problem using a "meta-model". The examples given for the inductive biases seem to be simply relations that exist between various hyper-parameters and design choices in the optimization domain. For instance, the example given "using a large number of layers without batch normalization degrades the performance” seems like a simple relation that can be learned by any model-based AutoML framework (even BO can in principle learn this). Why is the meta-GNN a "meta" model - does it learn anything across different tasks or datasets? The task agnostic component is not really learned across different tasks as far as I can tell. It is more like a component that captures some domain-specific knowledge about the design space. The meta-learning terminology is confusing to the reader.
- It is stated that other sample-based AutoML methods require "thousands of models from scratch". This may be somewhat true, but there also exist efficient methods like Successive Halving and Hyperband which train many models but using a small fraction of the training data in order to eliminate designs that are not promising very quickly. Why were these methods not discussed or used as baselines?
- The main algorithm is not described in sufficient detail to allow the reader to follow what is going on. Algorithm 1 is presented with little-to-no discussion of any of the subroutines that are present. Figure 1(b) is not sufficiently explained in the text. Important details like the design distance are offloaded to the appendix with little discussion.

Other questions:
- Section 4.2: what are the "one-shot methods"?
- Table 1 and Table 2: what statistical test was used to compute p-values + why are the p-values missing from some columns?
- Section 3.2: can you please elaborate on how the "instance-wise performances" are measured? In Figure 2 they seem to be binary vectors. What exactly do these vectors represent?
- Section 2: what is a "computational cell"?


**Summary Of The Paper:**

The paper describes a new method for AutoML (FALCON) that comprises a online search in the space of designs that is guided by a GNN model ("meta-GNN") that is trained along the way. The GNN comprises a task-agnostic component that tries to capture similarity between designs using various features and relations, as well as a task-specific component that performs label propagation based on a set of evaluated designs, and is trained using a pairwise rank loss. Experiments are presented on a variety of graph-based AutoML tasks, and results on an image classification task are included in the Appendix.

**Summary Of The Review:**

This paper contains some interesting work and ideas and the results are promising, but needs a rewrite to clean up some of the terminology used and present the details of the algorithm in a clearer way.

---

> ### Author Response · Authors · 2022-11-13
> **Reply to Reviewer ZhyY**
>
> We appreciate your comments! To address your concerns, we prudently clarify the terminology and add more details.
>
> **1. Clarification on the terminology of inductive biases**
>
> We would like to clarify the terminology of inductive biases here.
>
> The inductive biases refer to effects of multiple variables interacting together. We call them inductive biases because such relations can happen in multiple parts of the design space.
> (refer to the experiment where we find a trend in multiple parts of the design space)
> BO would require sampling of these values at every single occurrence (there could be exponentially many of these due to combinations of other design variables). This is simply not practical. And we have elaborated the concept more in the revision, which is highlighted in the Introduction Section.
>
> **2. Clarification on the terminology of meta-model and task-agnostic component**
>
> We use “meta-” because our meta-GNN takes model designs as input rather than raw data. In other words, it is built on top of the candidate models. For the task-agnostic module, it is applicable to any datasets (different downstream tasks) where the design space is unchanged. For example, the design space defined in Table 4 and Table 5 don’t rely on dataset.
>
> **3. Discussion about the exist efficient methods like Successive Halving and Hyperband**
>
> Thanks for bringing these works to us! We have added and discussed them in the related work (highlighted in the text).
>
> Specifically, these bandit-based hyperparameter search methods (as you mentioned) aim to reduce the computational cost. For example, Successive Halving [1] allocates the training resources to more potentially valuable models based on the early stage training information. Hyperband [2] further extends it via using different budgets to find the best configurations to avoid the trade-off between selecting the configuration number and allocating the budget. However, their selective mechanisms are only based on the model performance and lack of deep knowledge, which draws less insight into the relation of design variables and limits the sample efficiency.
>
> **4. Details about the main algorithm**
>
> Thanks for this comment. In the search strategy of Section 3.3, we described the Initialization, Meta-GNN training, and Exploration via inference which are corresponding to each main step in Algorithm 1. We have revised the statements about the main algorithm for better clarity. Specifically, we detail the steps in Algorithm 1 by adding more elaborations and more statements on Section 3.3.
>
> For the discussion of Figure 1 (b), please see its caption and the last paragraph in Section 3.3 (we move the discussion here for your convenience):
>
> “
> At every iteration, FALCON extends the design subgraph through the last two steps. After several iterations, it selects and retrains a few designs in the search trajectory with top performances. Overall, FALCON approaches the optimal design navigated by the relational inductive bias learned by meta-GNN, as shown in Figure 1 (b).
> ”
>
> Thus, Figure 1 (b) is an overview and intuition about our search strategy. We emphasize that there are too many important parts required by the review. It simply has to be the case that some parts are off-loaded to the appendix. Every single line we decide to place in the main text is crucial to our understanding. We made a conscious effort in making the appendix to be easy to refer to, by pointing out the exact position in the appendix where each discussion takes place.
>
>
>
>
> **5. The definition of one-shot methods**
>
> In the second paragraph of Related Work, we have discussed a series of works in the scope of one-shot AutoML methods. This terminology comes from the feature that these methods leverage weight sharing among designs in search space to train a supernet. For example, one can have, say, five different choices (GCN, GIN, Pooling, etc.) for a layer in the neural network. Instead of training five separate models, one-shot methods train a single model containing all five operations. Please refer to previous works [3, 4] for more details.
>
> **6. Why some columns are without p-values and the statistical test used to compute p-values**
>
> Please see the caption on Table 1 where we state “We compute p-value on our method with the best AutoML baselines”. Thus, for the Cora (Table 1) and AIDS (Table 2)  datasets, we didn’t compute p-value since FALCON is not outperforming all the baselines and we only computed for the datasets where FALCON output the best performance. For Table 1, we only compute the p-value for the final task performance, since the results of time costs are more deterministic with the computational budget (i.e., we set the number of exploration sizes).
>
> Specifically, we perform a T-test to compute p-values. We have updated our paper to include this information for better clarification.

---

> > ### Author Response · Authors · 2022-11-13
> > **(Cont.) Reply to Reviewer ZhyY**
> >
> >
> > **7. Instance-wise performances**
> >
> > Please see the first sentence in the “Task-specific module” paragraph. The instance-wise performance is the design performance on each selected training instance. In the case of Figure 2, we use accuracy as the metric, thus the $(i, j)$ element of the top left matrix represents whether the $i$th design can correctly predict the label of instance $j$. We have modified our paper to make it more clear. This information sheds light into the properties of each design toward the selected instances, and is shown to improve the expressiveness of our meta-GNN.
> >
> > **8. The definition of computational cell**
> >
> > Generally, a computational cell is a unit / building block of a computational graph representing the computing process of a neural architecture. Please refer to [5] for more details.
> >
> > --------------------------------
> >
> > We believe the concerns regarding clarity can all be answered by referring to the text. We would like to thank the reviewer and we also elaborate the definition of inductive biases to the main text (see revision) and add more details of algorithm. We genuinely hope our responses can address all of your concerns and please let us know if there are any remaining questions!
> >
> >
> > -----------------
> >
> > **Reference**:
> >
> > [1] Zohar Shay Karnin, Tomer Koren, and Oren Somekh. Almost optimal exploration in multi-armed bandits. In ICML, 2013.
> >
> > [2] Lisha Li, Kevin G. Jamieson, Giulia DeSalvo, Afshin Rostamizadeh, and Ameet Talwalkar. Hyperband: A novel bandit-based approach to hyperparameter optimization. J. Mach. Learn. Res., 2017.
> >
> > [3] Bender, G., Kindermans, P.J., Zoph, B., Vasudevan, V., Le, Q. Understanding and simplifying one-shot architecture search. International Conference on Machine Learning 2018.
> >
> > [4] Brock, A., Lim, T., Ritchie, J.M., Weston, N. Smash: one-shot model architecture search through hypernetworks. arXiv preprint arXiv:1708.05344
> >
> > [5] Hieu Pham,  Melody Y. Guan,  Barret Zoph,  Quoc V. Le,  and Jeff Dean.   Efficient neural architecture search via parameter sharing. In ICML, 2018.

---

> ### Author Response · Authors · 2022-11-18
> **We're looking forward to your reply!**
>
> Dear Reviewer ZhyY,
>
>
> As the discussion phase is approaching the end, we would be very excited to hear more from you.
>
> We genuinely hope our responses can address all of your concerns and please let us know if there are any remaining questions!
>
> Thank you again for your time and efforts!
>
> Best,
>
> Authors

---

### Official Review · Reviewer_Wvmw · 2022-10-27

**Confidence:** 4
**Correctness:** 3
**Technical Novelty And Significance:** 2
**Empirical Novelty And Significance:** 3
**Recommendation:** 5

**Clarity, Quality, Novelty And Reproducibility:**

- Clarity: The paper is clearly written.

- Quality: The technical quality is good and the author also attached the source code.

- Novelty: The novelty of the ranking loss (equation 2) is marginal because using paired data for training a ranking model has been studied in "[NeurIPS2020] BRP-NAS: Prediction-based NAS using GCNs". However, I think using a design graph to accelerate the model choice search for GNN is novel.

- Reproducibility: The author has attached the source code in https://anonymous.4open.science/r/Falcon . The reviewer hasn't run the source code but believe it should be largely reproducible.


**Strength And Weaknesses:**

Strength

- The performance of FALCON on GNN tasks is strong.
- It is reasonable to view the design space as a graph. However, the final performance might be sensitive to the rules that FALCON pick for constructing the graphs.


Weaknesses

- The author only conducted experiments on GNN tasks. Experiments on Image tasks are just conducted on a CIFAR-10 while previous papers usually report the performance on NAS-Bench. Actually, the author claimed in the end of Introduction (Page 2) that "Without loss of generality, we mainly focus on AutoML for graph representation learning in this paper.". I do not think that the author can claim "Without loss of generality" by just conducting experiments on GNNs. The method is very general and the author claimed it to be an AutoML framework. Experiments on a more diverse collection of tasks (such as more NAS-Bench, or NLP tasks) is necessary.
- The method is largely based on inductive bias of smoothness. However, I think smoothness won't hold in the general scenario (e.g., certain learning rates are preferred for certain network architectures). Thus, the author may need to elaborate more on the smoothness hypothesis.
- I think the method is also related to Population-based Training (PBT), and the author needs to mention the connection / difference between FALCON and PBT.


**Summary Of The Paper:**

The paper proposed an efficient automated model design algorithm called FALCON. FALCON views the overall design space as a design graph. Each node in the design graph represents an individual modeling choice, which includes the model architecture and the optimization hyper-parameters such as learning rate and weight decay. The nodes are connected following the predefined rules. The key intuition of FALCON is that the connected nodes tend to have the same performance. Thus, FALCON trains a performance model by applying GCN on the design graph, and use the performance model to explore new modeling choices and predict their performances. Results show that FALCON achieves better performance than previous auto-search algorithms, and the overall GPU-cost is not very huge. The author also conducted ablation analysis to show that modeling design relations is important for performance improvement.


**Summary Of The Review:**

I voted for weak reject due to concerns that the author is overselling their method. In addition, the smoothness inductive bias has not been discussed in depth.

---

> ### Author Response · Authors · 2022-11-13
> **Reply to Reviewer Wvmw**
>
> We are grateful for your comments and great suggestions! Here we provide point-to-point responses for your concerns.
>
> **1. Generality of our methods**
>
> The central concern of the reviewer is that there might be overselling of the method in terms of applicability to CV domains.
>
> In the revision, we therefore renamed the title to “Efficient Automatic Graph Learning via Design Relations” (Our title on the openreview can not be modified currently). We removed discussion of AutoML in CV domains in the abstract and  introduction, and instead discussed its potential applicability and CV application of Falcon as potential extensions.
>
> We believe our contribution to GNN Automl is significant regardless of the removal. Due to space limitation, the adaptation to general AutoML will be a future work.
>
> **2. Smoothness hypothesis**
>
> Thanks for this question! The smoothness hypothesis is basically the core assumption of label propagation. However, GNN is more expressive than a label/feature propagation on graph, therefore it has the capacity of picking up specific preferred hyperparameter values instead of choosing designs solely based on smoothness.
>
> Empirically, we further look into the predictions of GNN that are not smooth to demonstrate that some form of heterophily can be learned by expressive GNNs. Specifically, we use the following metric, where $\mathcal{N(i)}$ is the set of neighbors of node $i$, $\hat{Y_i}$ is the predicted performance of design $i$. Intuitively, it evaluates the extent that our model respects the smoothness.
>
>
> $$D\left(Y, \mathcal{G}\right)=\frac{1}{  |\mathcal{N}|} \sum_{i} \frac{1}{|\mathcal{N}(i)|}\sum_{j\in\mathcal{N}(i)}|\hat{Y_i}-\hat{Y_j}|  $$
>
> We compute this metric on the design subgraph at the last step of exploration instead of the whole design graph, which avoids accounting the relatively random prediction on nodes that the model has little information. The results on Cora (node classification dataset) and ER-MD (graph classification dataset) are shown as follows.
>
> ||   Cora 	| 	ER-MD	|
> |:----------:|:----------:|:----------:|
> |$D\left(Y, \mathcal{G}_s\right)$| 3.67\%  | 6.78% |
> |$D\left(Y,  \mathcal{G}_s\right)@[.5,.95]$| 3.15% | 4.33% |
>
>
> where $D\left(Y,  \mathcal{G}_s\right)@[.5,.95]$ only computes the nodes with the value $\frac{1}{|\mathcal{N}(i)|}\sum _{j\in \mathcal{N}(i) }|\hat{Y}_i-\hat{Y}_j|$ within the top 5% to 95% (to exclude outlier values). The results validate our statements that expressive  GNN is more expressive than label/feature propagation and can learn heterophily that goes beyond the smoothness hypothesis .
>
>
>
>
>
> **3. Related work: Population-based Training/ (PBT) [1]**
>
> Thank you for bringing this related research topic to us! Here we discuss the connection / difference between FALCON and PBT from various dimensions:
>
> |			|		|
> |:--------------------:|:-------------------------|
> |	Connection  	| **(1) Motivation**: FALCON and PBT are both (partly) motivated by the fact that search methods such as random or grid search consumes large amounts of computational resources.**(2) General Intuition**:  The intuitions behind FALCON and PBT are both to utilize the information on a subset of explored designs to conduct the search.|
> |	Difference 	| **(1) Applicability**: The experiments of PBT are mainly focused on searching hyperparameters, while FALCON can be applied to more general design space including architecture variables. **(2) Methodology**: FALCON is a sample-based algorithm which conducts sequential sampling from the unexplored designs, while PBT is a asynchronous optimisation algorithm which bridges parallel search methods and sequential optimisation methods.**(3) Techniques**: FALCON proposes the concept of design graph and uses meta-GNN, a metamodel over the design graph to conduct search, while PBT performs meta-optimisation using the collection of partial solutions in the population (candidate designs).|
>
> We have updated our paper to include the discussion and comparison. The concerns are clear and we genuinely hope our responses can address all of them and please let us know if there are any remaining questions!
>
>
> [1] Max Jaderberg, Valentin Dalibard, Simon Osindero, Wojciech M. Czarnecki, Jeff Donahue, Ali Razavi, Oriol Vinyals, Tim Green, Iain Dunning, Karen Simonyan, Chrisantha Fernando, Koray Kavukcuoglu. Population Based Training of Neural Networks.

---

> > ### Comment · Reviewer_Wvmw · 2022-12-07
> > **Thanks for the rebuttal.**
> >
> > I still won't recommend for acceptance due to two reasons: 1) The author has not properly discussed the relationship between the proposed algorithm with prior work in hyper-parameter optimization. We can treat the network architecture as hyper-parameters and apply HPO algorithms for picking the best choice. Thus, methods like BOHB and PBT are also applicable. 2) The author changed the title from "Efficient Automatic Machine Learning via Design Graphs" to "Efficient Automatic Graph Learning via Design Relations". The algorithm is actually general and should not only be used for GNNs (this will largely limit the scope of the method). I do think the author should keep the original title, but conduct more experiments in traditional NAS benchmarks. This will make the paper much more solid.

---

> > > ### Author Response · Authors · 2022-12-11
> > > **Further reply**
> > >
> > > We thank reviewer Wvmw so much for the response! And we provide further reply as follows:
> > >
> > > **1. The relationship between the proposed algorithm with prior work in hyper-parameter optimization**
> > >
> > > Please see the related work in the revision where we discussed prior works in hyper-parameter optimization, including Successive Halving, Hyperband, and PBT. Specifically, we have
> > >
> > > "
> > > Successive Halving (Karnin et al., 2013) allocates the training resources to more potentially valuable models based on the early-stage training information. Li et al. (2017) further extend it using different budgets to find the best configurations to avoid the trade-off between selecting the configuration number and allocating the budget. Jaderberg et al. (2017) combine parallel search and sequential optimisation methods to conduct fast search. However, their selective mechanisms are only based on the model performance and lack of deep knowledge, which draws less insight into the relation of design variables and limits the sample efficiency.
> > > "
> > >
> > > Moreover, we also include some hyper-parameter optimization methods like Bayesian optimization as our baselines (c.f., Table 1).
> > >
> > > For BOHB, it combines Bayesian optimization and Hyperband for both strong performance and fast convergence. However, it requires the evaluations on subsets with small budgets to represent evaluations on the entire training set and doesn't explicitly consider the relations between the hyper-parameters. To make our related works more comprehensive, we will add this work to the related work in the revision.
> > >
> > > **2. Applicability of our method**
> > >
> > > We believe Graph AutoML is a broad research area where we have shown promising improvements over the state-of-the-art Graph AutoML methods across 27 graph datasets. We believe the contributions of our work are sufficient currently. Moreover, the general design spaces of image networks (i.e., computational graphs) require more modeling techniques to measure the design similarities, which is out of the main focus of our work. We will extend our work to image domains and consider broader design spaces in future work.

---

> ### Author Response · Authors · 2022-11-18
> **We're looking forward to your reply!**
>
> Dear Reviewer Wvmw,
>
> As the discussion phase is approaching the end, we would be very excited to hear more from you.
>
> Your concerns are clear to us and we genuinely hope our responses can address all of them. Please let us know if there are any remaining questions!
>
> Thank you!
>
> Best,
>
> Authors

---

### Official Review · Reviewer_xdfg · 2022-10-30

**Confidence:** 4
**Correctness:** 3
**Technical Novelty And Significance:** 3
**Empirical Novelty And Significance:** 3
**Recommendation:** 8

**Clarity, Quality, Novelty And Reproducibility:**

The paper is very well-written. The overall quality of this work is very high.

**Strength And Weaknesses:**

**Strength**
1. The idea of design of modeling the search space as a design graph is quite novel. The method is also carefully designed with many thoughtful insights. For example, the construction of the design graph, the design distance, and the design of the subgraph. The task-agnostic and task-specific modules are also designed with caution.

2. The empirical evaluation is quite comprehensive and solid, showing improved performance compared to state-of-the-art baselines.

**Concerns and questions**

The scope of applicability of the proposed method is not clearly stated. It seems to me this method only works for cases where the choices are all discrete. What if there are hyperparameters with continuous domains? For example, the dropout ratio, momentum, and weight decay are all hyperparameters with continuous values. Is FALCON directly applicable to handling a search space with those continuous hyperparameters? The current experiment only takes several discrete values. Does it mean one has to discretize the continuous hyperparameters? If so, how to discrete them? In Table 12, the candidate values for the momentum of SGD are 0.5, 0.9, 0.99; the LR decay factor is 0.1, 0.2, 0.5, 0.8. I do not see any pattern in this discretization step. This specialized design space raises my doubts about how general this method can be. If there is a newly developed model needs to be tuned, how should one correctly configure those carefully designed components?

**Summary Of The Paper:**

This work proposes a sample-based method named FALCON, to search for the optimal model design. The key idea is to build a design graph over the design space of the architectures and hyperparameter choices. A meta-model is built to capture the relation between the design graph and model performance. The method uses GNN to learn and predict the performance of a specific design giving the corresponding nodes in the design graph. The GNN model includes both a task-specific module and a task-agnostic module.

**Summary Of The Review:**

This work proposes a sample-based method named FALCON, to search for the optimal model design. The key idea is to build a design graph over the design space of the architectures and hyperparameter choices. A meta-model is built to capture the relation between the design graph and model performance. The method uses GNN to learn and predict the performance of a specific design giving the corresponding nodes in the design graph. The GNN model includes both a task-specific module and a task-agnostic module.

The idea of design of modeling the search space as a design graph is quite novel. The method is also carefully designed with many thoughtful insights. For example, the construction of the design graph, the design distance, and the design of the subgraph. The task-agnostic and task-specific modules are also designed with caution. The empirical evaluation is quite comprehensive and solid, showing improved performance compared to state-of-the-art baselines.

But I do have concerns about the applicability of the proposed method to more general search spaces (see concerns and questions in the Strength And Weaknesses section)

---

> ### Author Response · Authors · 2022-11-13
> **Reply to Reviewer xdfg**
>
> Thank you for your time and insightful suggestions! We are very glad that you have a positive impression on the novelty and empirical evaluation of our work. According to your comments, we provide the responses as follows:
>
> **1. Hyperparameters with continuous domains (applicability in more general search space)**
>
> Thanks. In our experiments, we use some common hyperparameter values based on prior knowledge. For example, 0.9 and 0.99 are common values of SGD momentum.
>
> Besides that, for a newly developed model, one solution for discretization is sensitivity analysis. The intuition here is that we conduct a density discretization where the density of each hyperparameter is proportional to its sensitivity on model performance. Specifically, to customize the discrete design space for the new model, we first split continuous design choices into bins, e.g., [0, 0.25],... [0.75, 1.0] for LR decay factor. As a warm start, we could randomly sample a value for each bin and explore the corresponding design (train it for a few epochs). After that, we analyze the sensitivity of the continuous design variable in terms of the design performance and only select the representative value (e.g., median) in each bin with high sensitivity. This is a general approach to discretize hyperparameters with continuous domains and reduce the scale of the design graph.
>
> This question also gives us huge inspiration in adapting FALCON to continuous domains. We also provide a discussion on this point as follows.
>
>
> **2. Potential future direction in handling continuous domains**
>
> One potential future direction, in our minds, is to develop the design graph into a dynamic graph, where the candidate values in each design variable change during the search process. This could enable a more flexible adjustment, especially in the continuous domain. For example, suppose meta-GNN discovers that the model performance is highly correlated with weight decay, and there is a huge performance gap between weight decay=1e-3 and weight decay=1e-4. In that case, we can add a design choice between 1e-3 and 1e-4 to enable a more fine-grained search and better approximation when the design variables are in the continuous domain. We have updated our paper (c.f. the Future Work section) to include this future direction.

---

### Official Review · Reviewer_1F3M · 2022-11-04

**Confidence:** 2
**Correctness:** 1
**Technical Novelty And Significance:** 2
**Empirical Novelty And Significance:** 2
**Recommendation:** 3

**Clarity, Quality, Novelty And Reproducibility:**

The novelty of the main idea seems questionable. When we look at the empirical setting of the work, we see that the design parameters enable a simple approach without any design graphs: each parameter can be quantified as a separate dimension of a joint design space, and the problem becomes a black-box optimization on the obtained grid. Why this scenario is not addressed in the paper?

Some parts of the paper lacks of details. In particular, GNN meta-model and its training pipeline is not fully described The pipeline of the empirical part is not reproducible based solely on the text of the paper. The whole system is complex, it has many hyperparameters and itself design choices. This choice is not transparent, and I cannot be sure that the obtained empirical improvements are not an attribute of these choices.

**Strength And Weaknesses:**

+ The idea of representing the design space as a graph looks novel for NAS

- Related work seems incomplete. In particular, the claim “Previous works generally consider each design choice as isolated from other designs” seems too strong. I think, there should be an overview of previous papers related to meta-models over design spaces. Apart from NER, there should be also literature on design spaces in the context of classical ML. In this context, I am not convinced that the scientific problem is well-formulated. When we look at the empirical setting of the work, we see that the design parameters enable a simpler approach without any design graphs: each parameter can be quantified as a separate dimension of a joint design space, and the problem becomes a black-box optimization on the obtained grid.
- The motivation behind the proposed Falcon is questionable. Why not using Bayesian approaches? Gaussian processes?
- Some parts of the paper lacks of details. In particular, GNN meta-model and its training pipeline is not fully described
- The pipeline of the empirical part is not reproducible based solely on the text of the paper. The whole system is complex, it has many hyperparameters and itself design choices. This choice is not transparent, and I cannot be sure that the obtained empirical improvements are not an attribute of these choices.

**Summary Of The Paper:**

In this paper, a new method of sample-based NAS is proposed, Falcon. The key idea is to represent the model design space as a graph, where nodes correspond to separate designs and an edge corresponds to a minimal change in one of the design parameters. The problem of architecture search is thus translated to a black-box optimization task over the design graph. Falcon builds a GNN meta-model, which approximates the values in the nodes and use it in a search strategy.

**Summary Of The Review:**

I think, this paper does not appropriately place itself in the context of state-of-the-art and classical algorithms relevant to the problem, and needs a major revision.

---

> ### Author Response · Authors · 2022-11-13
> **Reply to Reviewer 1F3M**
>
> We appreciate your comments! However, we believe there are significant misunderstandings regarding our work, and here are our clarifications:
>
> **1. “Related work should cover previous papers related to meta-models over design spaces”**
>
> 1) To the best of our knowledge, we are the first work to explicitly consider the relations between model designs, and build a meta-graph on the design space.
>
> 2) We did discuss papers related to other notions of meta-model. For example, we discussed Graph HyperNetwork (please see “Graph-based AutoML methods” paragraph in related works) and one-shot AutoML methods which involve training a supernet representing the design space (please see “One-shot AutoML methods” paragraph in related works).
>
> 3) In the revision, we also include recent work about automatic graph learning, AutoGML, another meta-model design that uses a meta-learning approach to estimate the relevance of models to different graphs. We have uploaded the revision to include this additional work.
>
> **We believe that the related works are sufficient to reflect a wide perception of this field and our problem setting.**
>
> **2. “Apart from NER, there should also be literature on design spaces in the context of classical ML”**
>
> Could you please explain what “NER” is? Obviously this should not be related to named entity recognition. Perhaps you mean NAS? But as mentioned, the related works in NAS, design space (see discussion of the GNN design space paper), and meta-models are all been extensively discussed already.
>
>
> **3. “The design parameters enable a simpler approach without any design graphs, …,, and the problem becomes a black-box optimization on the obtained grid”**
>
> There is a **fundamental misunderstanding** here.
>
> 1) As mentioned in experiments, the simple approach mentioned by the reviewer is simply the **“Falcon-G”** ablation model (see the explanations in the Ablation model paragraph in the Experiment section). This means that we not only considered this simple approach but also demonstrated superiority through experiments (as seen by the p-value of the results). Furthermore, we theoretically justified the soundness of our approach compared to the simple approach, that Falcon considers dependencies between multiple design variables.
>
> 2) Note that our method can be adapted to any design space beyond the grid, and we did demonstrate such cases in our work. Please see Appendix A.2.1 (Page 15), where we demonstrate that the design graph constructed under dependency rules is more complex. Specifically, in Table 5, the number of message-passing layers and node pooling layers are dependent. This also shows the applicability of FALCON in various design spaces.
>
> 3) We believe the introduction of the meta-GNN (black-box model) is well motivated. As we stated clearly in Section 3.1 “We introduce the construction of design graph, and formulate the AutoML goal as a search on the design graph for the node with the best task performance”, and thus we introduce and leverage meta-GNN to perform fast inference on the design subgraph in Section 3.2 and 3.3.
>
> Therefore we prudently argue this main concern is incorrect due to the misunderstanding.
>
>
> **4. The motivation behind the proposed Falcon…Why not using Bayesian approaches? Gaussian processes?**
>
> We respectfully argue that our method is well motivated and superior to Bayesian approaches (which we also compared in experiments) based on the following statements and experiments in our work. Most justifications are already mentioned in the paper and we have to refer to the parts of the paper that can answer the reviewer’s concern.
>
> 1) In the **Introduction section**, we already emphasized that the key motivation of our work is the explicit modeling of design relations, while Bayesian and Gaussian Process approaches fail to do it.
>
> 2) In the **Related Work section**, we have stated that the sample-based approaches, including Bayesian approaches, usually involve sampling a large number of designs, thus having low sample efficiency. Also, we have considered task-specific information besides the design features, while Bayesian approaches and Gaussian processes don’t include such information.
>
> 3) In Figure 3 (the **Experiments section**), we have empirically shown that Bayesian Optimization performs badly even with a large number of explored nodes. And we attribute such failure to its inability to deal with high-dimensional design features.

---

> > ### Author Response · Authors · 2022-11-13
> > **(Cont.) Reply to Reviewer 1F3M**
> >
> >
> > **5. “GNN meta-model and its training pipeline is not fully described”**
> >
> >
> > For **meta-GNN**, please see Figure 2, which shows the framework of our meta-GNN. In Section 3.2, we described each component in detail, including the task-agnostic and task-specific model. For task-agnostic models, we clearly described the design and relational encoders as well as the message-passing module. Also, we described each step of the task-specific model. Finally, we proposed the objective of meta-GNN via Eq (2).
> >
> > For the **training pipeline** of our method, we have described it in Algorithm 1, where each major step is explained and corresponds to Section 3.3. Moreover, we added more notes on Algorithm 1 and more statements on Section 3.3 to improve the clarity.
> >
> > Furthermore, please see Appendix A for more training details.
> >
> >
> > **6.“The whole system is complex,”**
> >
> > We believe each component in our framework has a clear purpose. For example, the task-agnostic module is used to capture the information on design features solely, and label propagation aims to infer the performance distribution of other unknown designs (clearly explained in the method section and ablation studies).
> >
> > Even the code is fully released (see links). It is thus clear that not being able to understand the pipeline in a quick look does not imply that the model is not comprehensively described and justified.
> >
> > This logic simply does not apply to research papers, where most methods introduced have some form of complexity.
> >
> > **7. “It has many hyperparameters and design choices. This choice is not transparent.”**
> >
> > Here are the major hyperparameters in our modeling:
> >
> > 1) **Optimization hyperparameters**: Trade-off hyper-parameter $\lambda$ and temperature $\tau$ in the training objective which are both set as 0.1 in all our experiments, and the learning rate in the optimization process.
> >
> > 2) **Modeling hyperparameters**:  $\alpha$ in Equation 1, which is set as 0.9 in all our experiments, number of hops in message-passing, and number of random nodes.
> >
> > The values of hyperparameters that are fixed in all our experiments are included in our code. Besides the fixed hyperparameters, the learning rate is tuned to ensure the loss decreases properly, and we have studied the sensitivity of the number of hops and random nodes in Appendix D (we mentioned it in Section 4.2). From the results of sensitivity analysis, we found that FALCON outperforms the best AutoML baselines with a large range of hyper-parameters. Specifically, 73% and 97% hyper-parameter combinations of FALCON rank best among the baselines in CiteSeer and Tox21-AhR, respectively. This validates the effectiveness of our method, considering different combinations of hyperparameters.
> >
> > For design choices, they are part of a predefined search problem and are independent of the proposed method since our method is applicable to any design choices (design space).
> >
> >
> > --------------------------
> >
> > Overall, we thank the reviewer again for the comments, and we provide the responses that point to parts of the paper that exactly address the concern of the reviewer. We hope the reviewer could adjust the score after the misunderstanding is resolved.

---

> ### Author Response · Authors · 2022-11-18
> **We're looking forward to your reply!**
>
> Dear Reviewer 1F3M,
>
>
> As the discussion phase is approaching the end, we would be very excited to hear more from you.
>
> We have provided the responses about your concerns. And we also welcome more questions and discussions!
>
> Thank you so very much!
>
> Best,
>
> Authors

---

### Author Response · Authors · 2022-11-13
**General Response**

We sincerely appreciate all reviewers' time and efforts in reviewing our paper! Evidently you spent a substantial time reviewing our paper. We would like to thank all reviewers for providing many insightful and valuable suggestions. Here is a summary of our main updates:

**More Related Works**: we add more related works including (1) AutoGML (1F3M), a paper related to meta-models, (2) Successive Halving and Hyperband (ZhyY), two efficiency-oriented hyper-parameter search methods (3) Population-based Training (Wvmw) which combines parallel search and sequential optimisation.

**Clarification**: We clarify our motivation (1F3M), hypothesis (Wvmw) and search algorithm (1F3M, ZhyY). We have updated our paper for better clarity.

**Future work**: We discuss a potential future direction in handling continuous domains inspired by the comments of reviewer xdfg.

We've highlighted the updates in the revision. We hope our responses can clarify all your confusion and alleviate all concerns. We thank all reviewers again. Looking forward to your reply!

---

> ### Author Response · Authors · 2022-11-15
> **We hope that the reviewers find the clarification useful and let us know if the issues have been addressed**
>
> We thank all reviewers for their time during the challenging 2 week discussion period. We hope that the reviewers find the clarification useful and let us know if the issues have been addressed. We hope that the reviewers could raise the score if the corresponding issues have been addressed.

---

### Decision · Program_Chairs · 2023-01-20

**Decision:**

Reject

**Justification For Why Not Higher Score:**

Limited applicability of the approach, thus limited significance.

Clarity issues in the presentation.

**Justification For Why Not Lower Score:**

Can't be lower.

**Metareview: Summary, Strengths And Weaknesses:**

This paper presents an approach for AutoML in the domain of optimizing hyperparameters for graph neural networks.  The main idea is to use a graph structure to represent the space of hyperparameters such that a GNN model can be used to exploit the structure of this space and make the search more efficient.

This paper is a bit below the bar of acceptance for a few reasons, but mostly:
1. The applicability and the empirical results presented in this paper is a bit limited, it only considers hyperparameters with small numbers of discrete choices, and with a heavy focus on the narrower GNN domain.  This restriction hurts the significance of this paper. The approach itself is general, so there’s no real reason for the authors to not do more in the more general setting, otherwise the audience needs to be convinced that the graph setting is somewhat special and people should care about this speciality.
2. Clarity of the approach can be improved, with multiple reviewers confused about the presentation.